# Throughput-Optimal Topology Design for Cross-Silo Federated Learning

**Othmane Marfoq**
Inria, Université Côte d'Azur,
Accenture Labs,
Sophia Antipolis, France
othmane.marfoq@inria.fr

**Chuan Xu**
Inria, Université Côte d'Azur,
Sophia Antipolis, France
chuan.xu@inria.fr

**Giovanni Neglia**
Inria, Université Côte d'Azur,
Sophia Antipolis, France
giovanni.neglia@inria.fr

**Richard Vidal**
Accenture Labs,
Sophia Antipolis, France
richard.vidal@accenture.com

## Abstract

Federated learning usually employs a server-client architecture where an orchestrator iteratively aggregates model updates from remote clients and pushes them back a refined model. This approach may be inefficient in cross-silo settings, as close-by data silos with high-speed access links may exchange information faster than with the orchestrator, and the orchestrator may become a communication bottleneck. In this paper we define the problem of topology design for cross-silo federated learning using the theory of max-plus linear systems to compute the system throughput—number of communication rounds per time unit. We also propose practical algorithms that, under the knowledge of measurable network characteristics, find a topology with the largest throughput or with provable throughput guarantees. In realistic Internet networks with 10 Gbps access links at silos, our algorithms speed up training by a factor 9 and 1.5 in comparison to the server-client architecture and to state-of-the-art MATCHA, respectively. Speedups are even larger with slower access links.

## 1  Introduction

Federated learning (FL) "*involves training statistical models over remote devices or siloed data centers, such as mobile phones or hospitals, while keeping data localized*" [56] because of privacy concerns or limited communication resources. The definition implicitly distinguishes two different settings [41]: the *cross-device* scenario including a large number (millions or even more) of unreliable mobile/edge devices with limited computing capabilities and slow Internet connections, and the *cross-silo* scenario with at most a few hundreds of reliable data silos with powerful computing resources and high-speed access links. While the first FL papers [72, 51] emphasized the cross-device setting, the cross-silo scenario has become popular for distributed training among banks [107], hospitals [19, 93, 69], pharmaceutical labs [67], and manufacturers [74].

In federated learning, clients (e.g., mobile devices or whole organizations) usually train the model through an iterative procedure under the supervision of a central orchestrator, which, for example, decides to launch the training process and coordinates training advances. Often—e.g., in FedAvg [72], SCAFFOLD [45], and FedProx [57]—the orchestrator directly participates to the training, by aggregating clients' updates, generating a new model, and pushing it back to the clients. Hence, clients only communicate with a potentially far-away (e.g., in another continent) orchestrator and do not

exploit communication opportunities with close-by clients. This choice is justified in the cross-device setting, where inter-device communication is unreliable (devices may drop-out from training at any time) and slow (a message needs to traverse two slow access links). But in the cross-silo setting, data silos (e.g., data centers) are almost always available, enjoy high-speed connectivity comparable to the orchestrator's one, and may exchange information faster with some other silos than with the orchestrator. An orchestrator-centered communication topology is then potentially inefficient, because it ignores fast inter-silo communication opportunities and makes the orchestrator a candidate for congestion. A current trend [104, 18, 100, 95, 7, 49, 53] is then to replace communication with the orchestrator by peer-to-peer communications between individual silos, which perform local partial aggregations of model updates. We also consider this scenario and study how to design the communication topology.

The communication topology has two contrasting effects on training duration. First, a more connected topology leads to faster convergence in terms of iterations or communication rounds, as quantified by classic worst-case convergence bounds in terms of the spectral properties of the topology [75, 24, 89, 90, 103, 40]. Second, a more connected topology increases the duration of a communication round (e.g., it may cause network congestion), motivating the use of degree-bounded topologies where every client sends and receives a small number of messages at each round [5, 61]. Recent experimental and theoretical work suggests that, in practice, *the first effect has been over-estimated by classic worst-case convergence bounds*. Reference [79] partially explains the phenomenon and overviews theoretical results proving asymptotic topology-independence [61, 81, 5]. [50, Sect. 6.3] extends some of the conclusions in [79] to dynamic topologies and multiple local updates. Experimental evidence on image classification tasks ([79, Fig. 2], [66, Fig 20.], [61, Fig. 3]) and natural language processing tasks ([61, Figs. 13-16]) confirms this finding. Motivated by these observations, this paper focuses on the effect of topology on the duration of communication rounds.

Only a few studies have designed topologies taking into account the duration of a communication round. Under the simplistic assumption that the communication time is proportional to node degree, MATCHA [104] decomposes the set of possible communications into matchings (disjoint pairs of clients) and, at each communication round, randomly selects some matchings and allows their pairs to transmit. MATCHA chooses the matchings' selection probabilities in order to optimize the algebraic connectivity of the expected topology. Reference [78] studies how to select the degree of a regular topology when the duration of a communication round is determined by stragglers [44, 55]. Apart from these corner cases, "*how to design a [decentralized] model averaging policy that achieves the fastest convergence remains an open problem*" [41].

Our paper addresses this open problem. It uses the theory of linear systems in the max-plus algebra [6] to design cross-silo FL topologies that minimize the duration of communication rounds, or equivalently maximize the system *throughput*, i.e., the number of completed rounds per time unit. The theory holds for synchronous systems and has been successfully applied in other fields (e.g., manufacturing [16], communication networks [54], biology [12], railway systems [31], and road networks [25]). Synchronous optimization algorithms are often preferred for federated learning [9], because they enjoy stronger convergence guarantees than their asynchronous counterparts and can be easily combined with cryptographic secure aggregation protocols [8], differential privacy techniques [1], and model and update compression [111, 101, 88, 13].

To the best of our knowledge, this paper is the first work to take explicitly in consideration all delay components contributing to the total training time including computation times, link latencies, transmission times, and queueing delays. It complements the topology design approaches listed above that only account for congestion at access links [104] and straggler effect [78].

The algorithms we propose (Sect. 3) are either optimal or enjoy guaranteed approximation factors. Numerical results in Sect. 4 show significant training speed-up in realistic network settings; the slower the access links, the larger the speedups.

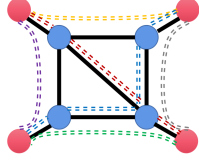 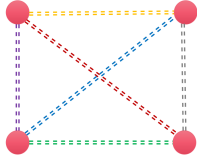 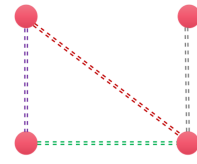

(a) Underlay $\mathcal{G}_u = (\mathcal{V} \cup \mathcal{V}', \mathcal{E}_u)$   (b) Connectivity graph $\mathcal{G}_c = (\mathcal{V}, \mathcal{E}_c)$   (c) Overlay $\mathcal{G}_o = (\mathcal{V}, \mathcal{E}_o)$

Figure 1: Examples for underlay, connectivity graph, and overlay, with routers (blue nodes), silos (red nodes), underlay links (solid black lines), and information exchanges (dashed lines).

## 2 Problem Formulation

### 2.1 Machine Learning Training

We consider a network of $N$ siloed data centers who collaboratively train a global machine learning model, solving the following optimization problem:

$$\underset{\boldsymbol{w} \in \mathbb{R}^d}{\text{minimize}} \sum_{i=1}^{N} p_i \mathbb{E}_{\xi_i} \left[ f_i(\boldsymbol{w}, \xi_i) \right], \tag{1}$$

where $f_i(\boldsymbol{w}, \xi_i)$ is the loss of model $\boldsymbol{w}$ at a sample $\xi_i$ drawn from data distribution at silo $i$ and the coefficient $p_i > 0$ specifies the relative importance of each silo, with two natural settings being $p_i$ equal to 1 or to the size of silo $i$'s local dataset [56]. In the rest of the paper we consider $p_i = 1$, but our analysis is not affected by the choice of $p_i$.

In order to solve Problem (1) in an FL scenario, silos do not share the local datasets, but periodically transmit model updates, and different distributed algorithms have been proposed [57, 72, 58, 45, 104, 52, 103]. In this paper we consider as archetype the decentralized periodic averaging stochastic gradient descent (DPASGD) [103], where silos are represented as vertices of a communication graph that we call *overlay*. Each silo $i$ maintains a local model $\boldsymbol{w}_i$ and performs $s$ mini-batch gradient updates before sending its model to a subset of silos $\mathcal{N}_i^-$ (its out-neighbours in the overlay). It then aggregates its model with those received by a (potentially different) set of silos $\mathcal{N}_i^+$ (its in-neighbours). Formally, the algorithm is described by the following equations:

$$\boldsymbol{w}_i(k+1) = \begin{cases} \sum_{j \in \mathcal{N}_i^+ \cup \{i\}} \boldsymbol{A}_{i,j} \boldsymbol{w}_j(k), & \text{if } k \equiv 0 \pmod{s+1}, \\ \boldsymbol{w}_i(k) - \alpha_k \frac{1}{m} \sum_{h=1}^{m} \nabla f_i \left( \boldsymbol{w}_i(k), \xi_i^{(h)}(k) \right), & \text{otherwise.} \end{cases} \tag{2}$$

where $m$ is the batch size, $\alpha_k > 0$ is a potentially varying learning rate, and $\boldsymbol{A} \in \mathbb{R}^{N \times N}$ is a matrix of non-negative weights, referred to as the *consensus matrix*. For particular choices of the matrix $\boldsymbol{A}$ and the number of local updates $s$, DPASGD reduces to other schemes previously proposed [61, 58, 110], including FedAvg [72], where the orchestrator just performs the averaging step (this corresponds to its local loss function $f_i(.)$ being a constant). Convergence of (2) was proved in [103].

In this paper we study how to design the overlay in order to minimize the training time. While we consider DPASGD, our results are applicable to any synchronous iterative algorithm where each silo alternates a local computation phase and a communication phase during which it needs to receive inputs from a given subset of silos before moving to the next computation phase. This includes the distributed algorithms already cited, as well as push-sum training schemes [5, 91, 87, 76, 23, 98, 109] and in general the *black-box optimization procedures* as defined in [90].

### 2.2 Underlay, Connectivity graph, and Overlay

FL silos are connected by a communication infrastructure (e.g., the Internet or some private network), which we call *underlay*. The underlay can be represented as a directed graph (digraph) $\mathcal{G}_u = (\mathcal{V} \cup \mathcal{V}', \mathcal{E}_u)$, where $\mathcal{V}$ denotes the set of silos, $\mathcal{V}'$ the set of other nodes (e.g., routers) in the network, and $\mathcal{E}_u$ the set of communication links. For simplicity, we consider that each silo $i \in \mathcal{V}$ is connected to the rest of the network through a single link $(i, i')$, where $i' \in \mathcal{V}'$, with uplink capacity $C_{\text{UP}}(i)$ and downlink capacity $C_{\text{DN}}(i)$. The example in Fig. 1 illustrates the underlay and the other concepts we are going to define.

The *connectivity graph* $\mathcal{G}_c = (\mathcal{V}, \mathcal{E}_c)$ captures the possible direct communications among silos. Often the connectivity graph is fully connected, but specific NAT or firewall configurations may prevent some pairs of silos to communicate. If $(i, j) \in \mathcal{E}_c$, $i$ can transmit its updated model to $j$. The message experiences a delay that is the sum of two contributions: 1) an end-to-end delay $l(i, j)$ accounting for link latencies, and queueing delays long the path, and 2) a term depending on the model size $M$ and the *available bandwidth*[1] $A(i, j)$ of the path. Each pair of silos $(i, j)$ can use probing packets [39, 84, 38] to measure end-to-end delays and available bandwidths and communicate them to the orchestrator, which then designs the topology. We assume that in the stable cross-silo setting these quantities do not vary or vary slowly, so that the topology is recomputed only occasionally, if at all.

The training algorithm in (2) does not need to use all potential connections. The orchestrator can select a connected subgraph of $\mathcal{G}_c$. We call such subgraph *overlay* and denote it by $\mathcal{G}_o = (\mathcal{V}, \mathcal{E}_o)$, where $\mathcal{E}_o \subset \mathcal{E}_c$. Only nodes directly connected in $\mathcal{G}_o$ will exchange messages. We can associate a delay to each link $(i, j) \in \mathcal{E}_o$, corresponding to the time interval between the beginning of a local computation at node $i$, and the receiving of $i$'s updated model by $j$:

$$d_o(i,j) = s \times T_c(i) + l(i,j) + \frac{M}{A(i,j)} = s \times T_c(i) + l(i,j) + \frac{M}{\min\left( \frac{C_{\mathrm{UP}}(i)}{|\mathcal{N}_i^-|}, \frac{C_{\mathrm{DN}}(j)}{|\mathcal{N}_j^+|}, A(i',j') \right)}, \quad (3)$$

where $T_c(i)$ denotes the time to compute one local update of the model. We also define $d_o(i,i) = s \times T_c(i)$. Equation (3) holds under the following assumptions. First, each silo $i$ uploads its model in parallel to its out-neighbours in $\mathcal{N}_i^-$ (with a rate at most $C_{\mathrm{UP}}(i)/|\mathcal{N}_i^-|$). Second, downloads at $j$ happen in parallel too. While messages from different in-neighbours may not arrive at the same time at $j$'s downlink, their transmissions are likely to partially overlap. Finally, different messages do not interfere significantly in the core network, where they are only a minor component of the total network traffic ($A(i',j')$ does not depend on $\mathcal{G}_o$).

Borrowing the terminology from P2P networks [71] we call a network *edge-capacitated* if access links delays can be neglected, otherwise we say that it is *node-capacitated*. While in cross-device FL the network is definitely node-capacitated, in cross-silo FL—the focus of our work—silos may be geo-distributed data centers or branches of a company and then have high-speed connections, so that neglecting access link delays may be an acceptable approximation.

Our model is more general than those considered in related work: [104] considers $d_o(i,j) = M \times |\mathcal{N}_i^-|/C_{\mathrm{UP}}(i)$ and [78] considers $d_o(i,j) = T_c(i)$ (but it accounts for random computation times).

## 2.3   Time per Communication Round (Cycle Time)

Let $t_i(k)$ denote the time at which worker $i$ starts computing $w_i((s+1)k+1)$ according to (2) with $t_i(0) = 0$. As $i$ needs to wait for the inputs $w_j((s+1)k)$ from its in-neighbours, the following recurrence relation holds

$$t_i(k+1) = \max_{j \in \mathcal{N}_i^+ \cup \{i\}} (t_j(k) + d_o(j,i)). \quad (4)$$

This set of relations generalizes the concept of a linear system in the max-plus algebra, where the max operator replaces the usual sum and the $+$ operator replaces the usual product. We refer the reader to [6] for the general theory of such systems and we present here only the key results for our analysis.

We call the time interval between $t_i(k)$ and $t_i(k+1)$ a *cycle*. The average cycle time for silo $i$ is defined as $\tau_i = \lim_{k \to \infty} t_i(k)/k$. The cycle time 1) does not depend on the specific silo (i.e., $\tau_i = \tau_j$) [6, Sect. 7.3.4], and 2) can be computed directly from the graph $\mathcal{G}_o$ [6, Thm. 3.23]. In fact:

$$\tau(\mathcal{G}_o) = \max_{\gamma} \frac{d_o(\gamma)}{|\gamma|}, \quad (5)$$

where $\gamma$ is a generic circuit, i.e., a path $(i_1, \ldots, i_p = i_1)$ where the initial node and the final node coincide, $|\gamma| = p$ is the length of the circuit, and $d_o(\gamma) = \sum_{k=1}^{p-1} d_o(i_k, i_{k+1})$ is the sum of delays

Table 1: Algorithms to design the overlay $\mathcal{G}_o$ from the connectivity graph $\mathcal{G}_c$.

| Network | Conditions | Algorithm | Complexity | Guarantees |
|---|---|---|---|---|
| Edge-capacitated | Undirected $\mathcal{G}_o$ | Prim's Algorithm [85] | $\mathcal{O}(|\mathcal{E}_c| + |\mathcal{V}|\log|\mathcal{V}|)$ | Optimal solution (Prop. 3.1) |
| Edge/Node-capacitated | Euclidean $\mathcal{G}_c$ | Christofides' Algorithm [73] | $\mathcal{O}(|\mathcal{V}|^2 \log|\mathcal{V}|)$ | $3N$-approximation (Prop. 3.3,3.6) |
| Node-capacitated | Euclidean $\mathcal{G}_c$ and undirected $\mathcal{G}_o$ | Algorithm 1 (Appendix D) | $\mathcal{O}(|\mathcal{E}_c||\mathcal{V}|\log|\mathcal{V}|)$ | 6-approximation (Prop. 3.5) |

on $\gamma$. A circuit $\gamma$ of $\mathcal{G}_o$ is called *critical* if $\tau(\mathcal{G}_o) = d_o(\gamma)/|\gamma|$. There exist algorithms with different complexity to compute the cycle time [46, 20].

The cycle time is a key performance metric for the system because the difference $|t_i(k) - \tau(\mathcal{G}_o) \times k|$ is bounded for all $k \geq 0$ so that, for large enough $k$, $t_i(k) \approx \tau(\mathcal{G}_o) \times k$. In particular, the inverse of the cycle time is the *throughput* of the system, i.e., the number of communication rounds per time unit. An overlay with minimal cycle time minimizes the time required for a given number of communication rounds. This observation leads to our optimization problem.

## 2.4 Optimization Problem

Given a connectivity graph $\mathcal{G}_c$, we want the overlay $\mathcal{G}_o$ to be a strong digraph (i.e., a strongly connected directed graph) with minimal cycle time. Formally, we define the following *Minimal Cycle Time* problem:

Minimal Cycle Time (MCT)
**Input:** A strong digraph $\mathcal{G}_c = (\mathcal{V}, \mathcal{E}_c)$, $\{C_{\text{UP}}(i), C_{\text{DN}}(j), l(i,j), A(i',j'), T_c(i), \forall(i,j) \in \mathcal{E}_c\}$.
**Output:** A strong spanning subdigraph of $\mathcal{G}_c$ with minimal cycle time.

Note that the input does not include detailed information about the underlay $\mathcal{G}_u$, but only information available or measurable at the silos (see Sect. 2.2). To the best of our knowledge, our paper is the first effort to study MCT. The closest problem considered in the literature is, for a given overlay, to select the largest delays that guarantee a minimum throughput [28, 21].

# 3 Theoretical Results and Algorithms

In this section we present complexity results for MCT and algorithms to design the optimal topology in different settings. Table 1 lists these algorithms, their time-complexity, and their guarantees. We note that in some cases we adapt known algorithms to solve MCT. All proofs and auxiliary lemmas are in Appendix E.

## 3.1 Edge-capacitated networks

Remember that we call a network edge-capacitated if access links delays can be neglected, as it is for example the case whenever $\frac{1}{N} \times \min\left(C_{\text{UP}}(i), C_{\text{DN}}(j)\right) \geq A(i',j')$ for each $(i,j) \in \mathcal{E}_c$. In this setting (3) becomes

$$d_o(i,j) = s \times T_c(i) + l(i,j) + \frac{M}{A(i',j')}, \tag{6}$$

and then the delay between two silos does not depend on the selected overlay $\mathcal{G}_o$.

FL algorithms often use an *undirected* overlay with symmetric communications, i.e., $(i,j) \in \mathcal{E}_o \Rightarrow (j,i) \in \mathcal{E}_o$. This is the case of centralized schemes, like FedAvg, but is also common for other consensus-based optimization schemes where the consensus matrix $\boldsymbol{A}$ is required to be doubly-stochastic [77, 87, 103]—a condition simpler to achieve when $\mathcal{G}_o$ is undirected.

When building an undirected overlay, we can restrict ourselves to consider trees as solutions of MCT (Lemma E.1). In fact, additional links can only increase the number of circuits and then increase the cycle time (see (5)). Moreover, we can prove that the overlay has simple critical circuits of the form $\gamma = (i,j,i)$, for which $d_o(\gamma)/|\gamma| = (d_o(i,j) + d_o(j,i))/2$ (Lemma E.2). Intuitively, if we progressively build a tree using the links in $\mathcal{G}_c$ with the smallest average of delays in the two directions, we obtain the overlay with minimal cycle time. This construction corresponds to finding a minimum weight spanning tree (MST) in an opportune undirected version of $\mathcal{G}_c$:

**Proposition 3.1.** *Consider an undirected weighted graph $\mathcal{G}_c^{(u)} = (\mathcal{V}, \mathcal{E}_c^{(u)})$, where $(i,j) \in \mathcal{E}_c^{(u)}$ iff $(i,j) \in \mathcal{E}_c$ and $(j,i) \in \mathcal{E}_c$ and where $(i,j) \in \mathcal{E}_c^{(u)}$ has weight $d_c^{(u)}(i,j) = (d_o(i,j) + d_o(j,i))/2$. A minimum weight spanning tree of $\mathcal{G}_c^{(u)}$ is a solution of MCT when $\mathcal{G}_c$ is edge-capacitated and $\mathcal{G}_o$ is required to be undirected.*

Prim's algorithm [85] is an efficient algorithm to find an MST with complexity $\mathcal{O}(|\mathcal{E}_c| + |\mathcal{V}| \log |\mathcal{V}|)$ and then suited for the usual cross-silo scenarios with at most a few hundred nodes [41].

We have pointed out a simple algorithm when the overlay is undirected, but directed overlays can have arbitrarily shorter cycle times than undirected ones even in simple settings where all links in the underlay are bidirectional with identical delays in the two directions (see Appendix C). Unfortunately, computing optimal directed overlays is NP-hard:

**Proposition 3.2.** *MCT is NP-hard even when $\mathcal{G}_c$ is a complete Euclidean edge-capacitated graph.*

We call a connectivity graph $\mathcal{G}_c$ *Euclidean* if its delays $d_c(i,j) \triangleq s \times T_c(i) + l(i,j) + M/A(i',j')$ are symmetric ($d_c(i,j) = d_c(j,i), \forall i,j \in \mathcal{V}$) and satisfy the triangle inequality ($d_c(i,j) \leq d_c(i,k) + d_c(k,j), \forall i,j,k \in \mathcal{V}$). These assumptions are roughly satisfied for geographically distant computing clusters with similar computation times, as the delay to transmit a message between two silos is roughly an affine function of the geodesic distance between them [32]. Under this condition MCT can be approximated:

**Proposition 3.3.** *Christofides' algorithm [73] is a $3N$-approximation algorithm for MCT when $\mathcal{G}_c$ is edge-capacitated and Euclidean.*

The result follows from Christofides' algorithm being a 1.5-approximation algorithm for the Travelling Salesman Problem [73], and our proof shows that a solution of the Travelling Salesman Problem provides a $2N$-approximation of MCT. Note that Christofides' algorithm finds *ring* topologies.

### 3.2 Node-capacitated networks

When silos do not enjoy high-speed connectivity, congestion at access links can become the dominant contribution to network delays, especially when one silo communicates with many others. Intuitively, in this setting, good overlays will exhibit small degrees.

If $\mathcal{G}_o$ is required to be undirected, MCT can be reduced from the problem of finding the minimum bottleneck spanning tree with bounded degree $\delta > 1$ ($\delta$-MBST for short),[2] which is NP-hard.

**Proposition 3.4.** *In node-capacitated networks MCT is NP-hard even when the overlay is required to be undirected.*

We propose Algorithm 1 (see Appendix D), which combines existing approximation algorithms for $\delta$-MBST on a particular graph built from $\mathcal{G}_c$.

**Proposition 3.5.** *Algorithm 1 is a 6-approximation algorithm for MCT when $\mathcal{G}_c$ is node-capacitated and Euclidean with $C_{UP}(i) \leq \min\left(\frac{C_{DN}(j)}{N}, A(i',j')\right)$, $\forall(i,j) \in \mathcal{E}_c$, and $\mathcal{G}_o$ is required to be undirected.*

Finding directed overlays is obviously an NP-hard problem also for node-capacitated networks. Christofides' algorithm holds its approximation factor also in this more general case:

**Proposition 3.6.** *Christofides' algorithm is a $3N$-approximation algorithm for MCT when $\mathcal{G}_c$ is node-capacitated and Euclidean.*

## 4 Numerical Experiments

We adapted PyTorch with the MPI backend to run DPASGD (see (2)) on a GPU cluster. We also developed a separate network simulator that takes as input an arbitrary underlay topology described in the Graph Modelling Language [36] and silos' computation times and calculates the time instants at which local models $\boldsymbol{w}_i(k)$ are computed according to (2) (Appendix F). While

Table 2: Datasets and Models. Mini-batch gradient computation time with NVIDIA Tesla P100.

| Dataset | Task | Samples (x $10^3$) | Batch Size | Model | Parameters (x $10^3$) | Model Size (Mbits) | Computation Time (ms) |
|---|---|---|---|---|---|---|---|
| Shakespeare [14, 72] | Next-Character Prediction | 4,226 | 512 | Stacked-GRU [17] | 840 | 3.23 | 389.6 |
| FEMNIST [14] | Image classification | 805 | 128 | 2-layers CNN | 1,207 | 4.62 | 4.6 |
| Sentiment140 [30] | Sentiment analysis | 1,600 | 512 | GloVe [82]+ LSTM [37] | 4,810 | 18.38 | 9.8 |
| iNaturalist [99] | Image classification | 450 | 16 | ResNet-18 [35] | 11,217 | 42.88 | 25.4 |

Table 3: iNaturalist training over different networks. 1 Gbps core links capacities, 10 Gbps access links capacities. One local computation step ($s = 1$).

| Network name | Silos | Links | Cycle time (ms) | | | | | Ring's training speed-up | |
|---|---|---|---|---|---|---|---|---|---|
| | | | STAR | MATCHA$^{(+)}$ | MST | $\delta$-MBST | RING | vs STAR | vs MATCHA$^{(+)}$ |
| Gaia [38] | 11 | 55 | 391 | 228 (228) | 138 | 138 | **118** | 2.65 | 1.54 (1.54) |
| AWS North America [96] | 22 | 231 | 288 | 124 (124) | 90 | 90 | **81** | 3.41 | 1.47 (1.47) |
| Géant [29] | 40 | 61 | 634 | 452 (106) | **101** | 101 | 109 | 4.85 | 3.46 (0.81) |
| Exodus [68] | 79 | 147 | 912 | 593 (142) | 145 | 145 | **103** | 8.78 | 5.71 (1.37) |
| Ebone [68] | 87 | 161 | 902 | 580 (123) | 122 | 122 | **95** | 8.83 | 6.09 (1.29) |

PyTorch trains the model as fast as the cluster permits, the network simulator reconstructs the real timeline on the considered underlay. The code is available at `https://github.com/omarfoq/communication-in-cross-silo-fl`.

We considered three real topologies from *Rocketfuel engine* [94] (Exodus and Ebone) and from *The Internet Topology Zoo* [48] (Géant), and two synthetic topologies (AWS North-America and Gaia) built from the geographical locations of AWS data centers [38, 96] (Table 3). These topologies have between 11 and 87 nodes located in the same continent with the exception of Gaia, which spans four continents. We considered that each node is connected to a geographically close silo by a symmetric access link. See Appendixes G and H for a detailed description of the experiments and additional results.

We evaluated our solutions on three standard federated datasets from LEAF [14] and on iNaturalist dataset [99] with geolocalized images from over 8,000 different species of plants and animals (Table 2). For LEAF datasets, we generated non-iid data distributions following the procedure in [57]. For iNaturalist we assigned half of the images uniformly at random and half to the closest silo obtaining local datasets different in size and in the species represented (Appendix G).

Table 3 shows the effect of 6 different overlays when training ResNet-18 over iNaturalist in networks with capacities equal to 1 Gbps and 10 Gbps for core links and access links, respectively.[3] These overlays are *(1)* the STAR, corresponding to the usual server-client setting, where the orchestrator (located at the node with the highest load centrality [11]) averages all models at each communication round, *(2)* a dynamic topology built from MATCHA starting from the connectivity graph, *(3)* one built starting from the underlay and denoted as MATCHA$^+$ (in both cases MATCHA's parameter $C_b$ equals 0.5 as in experiments in [104][4]), *(4)* the minimum spanning tree (MST) from Prop. 3.1, *(5)* the $\delta$-minimum bottleneck tree ($\delta$-MBST) from Prop. 3.5, and *(6)* the directed RING from Prop. 3.6. In this particular setting, $\delta$-MBST selects the same overlay as MST. The consensus matrix $A$ is selected according to the local-degree rule [62].[5]

The overlays found by our algorithms achieve a higher throughput (smaller cycle time) than the STAR (the server-client architecture) and, in most cases, than state-of-the-art MATCHA$^{(+)}$. [6] In particular,

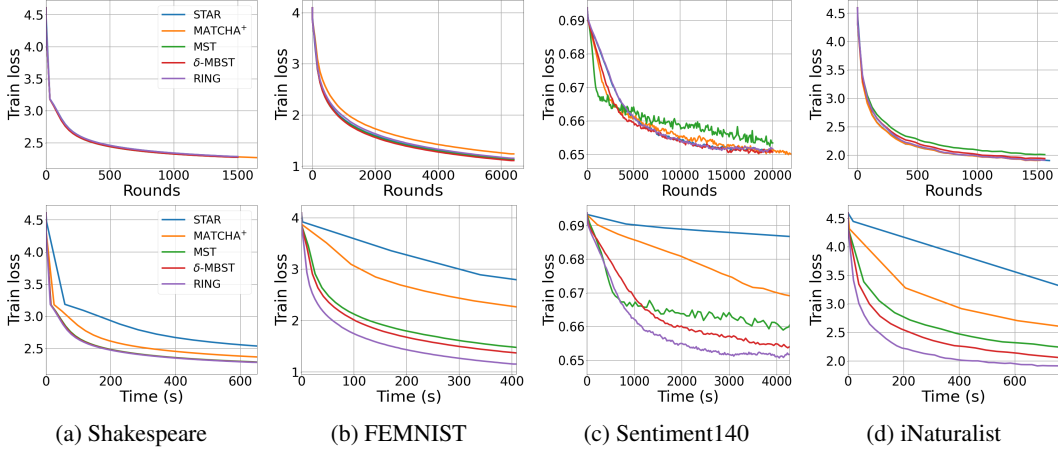

(a) Shakespeare      (b) FEMNIST      (c) Sentiment140      (d) iNaturalist

Figure 2: Effect of overlays on the convergence w.r.t. communication rounds (top row) and wall-clock time (bottom row) when training four different datasets on AWS North America underlay. 1 Gbps core links capacities, 100 Mbps access links capacities, $s = 1$.

the RING is between 3.3 ($\approx 391/118$ on Gaia) and 9.4 ($\approx 902/95$ on Ebone) times faster than the STAR and between 1.5 and 6 times faster than MATCHA. MATCHA$^+$ relies on the knowledge of the underlay—probably an unrealistic assumption in an Internet setting—while our algorithms only require information about the connectivity graph. Still, the RING is also faster than MATCHA$^+$ but on Géant network (where MST is the fastest overlay). From now on, we show only the results for MATCHA$^+$, as it outperforms MATCHA.

The final training time is the product of the cycle time and the number of communication rounds required to converge. The overlay also influences the number of communication rounds, with sparser overlays demanding more rounds [75, 24]. The last two columns in Table 3 show that this is a second order effect: the RING requires at most 20% more communication rounds than the STAR and then maintains almost the same relative performance in terms of the training time.[7] These results (and those in Fig. 2) confirm that the number of communication rounds to converge is weakly sensitive to the topology (as already observed in [61, 60, 49, 66] and partially explained in [86, 5, 79]). The conclusion is that overlays should indeed be designed for throughput improvement rather than to optimize their spectral properties: the topologies selected by our algorithms achieve faster training time than the STAR, which has optimal spectral properties, and MATCHA/MATCHA$^{(+)}$, which optimize spectral properties given a communication budget.

The same qualitative results hold for other datasets and Fig. 2 shows the training loss versus the number of communication rounds (top row) and versus time (bottom row) when training on AWS North America with 100 times slower access links. Other metrics for model evaluation (e.g., training/test accuracy) are shown in Appendix H.2. The advantage of designing the topology on the basis of the underlay characteristics is evident also in this setting.

Figure 3 illustrates the effect of access link speeds on the cycle time and the training time. When all silos have the same access link capacity (Fig. 3a), for capacity values smaller than 6 Gbps, the RING has the largest throughput followed by $\delta$-MBST, MST and MATCHA$^+$ almost paired, and finally the STAR. The advantage of topologies with small node degrees (like $\delta$-MBST and the RING) is someway expected in the small access link capacities regime, as the access link delay becomes the dominant term in (3). In particular, Eq. (5) and some simple calculations in Appendix B show that, with $N$ silos, the RING is up to $2N$ (=80 for Géant) times faster than the STAR and $C_b \times \max(\text{degree}(\mathcal{G}_u))$ (= 5 for Géant) times faster then MATCHA$^{(+)}$ for slow access links as confirmed in Fig. 3a (left plot). What is less expected (but aligned with our above observations about the importance to design overlays for throughput improvement) is that RING's throughput speedups lead to almost as large

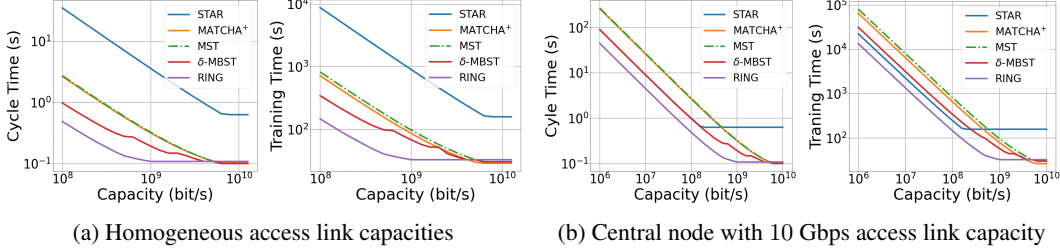

(a) Homogeneous access link capacities

(b) Central node with 10 Gbps access link capacity

Figure 3: Effect of access link capacities on the cycle time and the training time when training iNaturalist on Géant network. 1 Gbps core links capacities, $s = 1$. (3a): All access links have the same capacity. (3b): One node (the center of the star) has a fixed 10 Gbps access link capacity. The training time is the time when training accuracy reaches $55\%$.

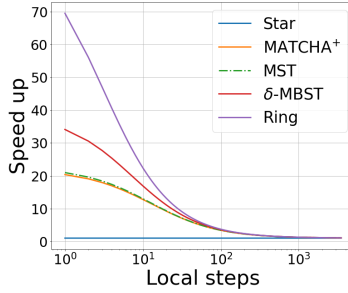

Figure 4: Throughput speedup in comparison to the STAR, when training iNaturalist over Exodus network. All links with 1 Gbps capacity.

training time speedups, even larger than those in Table 3: e.g. 72x in comparison to the STAR and 5.6x in comparison to MATCHA$^+$ for 100 Mbps access link capacities.

When the most central node (which is also the center of the STAR) maintains a fixed capacity value equal to 10 Gbps (Fig. 3b), the STAR performs better, but still is twice slower than the RING and only as fast as $\delta$-MBST. This result may appear surprising at first, but it is another consequence of Eq. (5) discussed in Appendix B. Again the relative performance of different overlays in terms of throughput is essentially maintained when looking at the final training time, with differences across topologies emerging only for those with very close throughputs, i.e., MST and MATCHA$^+$, and STAR and $\delta$-MBST in the heterogeneous setting of Fig. 3b.

When local computation requires less time than transmission of model updates, the silo may perform $s$ local computation steps before a communication round. As $s$ increases, the total computation time $(s \times T_c(i))$ becomes dominant in (3) and the throughput of different overlays become more and more similar (Fig. 4).[8] Too many local steps may degrade the quality of the final model, and how to tune $s$ is still an open research area [106, 105, 102, 64, 108, 50]. Our next research goal is to study this aspect in conjunction with topology design. Intuitively, a faster overlay reduces the number of local steps needed to amortize the communication cost and may lead to better models given the available time budget for training.

## 5   Conclusions

We used the theory of max-plus linear systems to propose topology design algorithms that can significantly speed-up federated learning training by maximizing the system throughput. Our results show that this approach is more promising than targeting topologies with the best spectral properties, as MATCHA$^{(+)}$ does. In future work, we will explore how to further speed-up training, e.g., by enriching the topologies found by our algorithms with additional links that improve connectivity without decreasing the throughput, and by carefully optimizing the weights of the consensus matrix.

# 6   Broader Impact

We have proposed topology design algorithms that can significantly speed-up federated learning in a cross-silo setting. Improving the efficiency of federated learning can foster its adoption, allowing different entities to share datasets that otherwise would not be available for training.

Federated learning is intended to protect data privacy, as the data is not collected at a single point. At the same time a federated learning system, as any Internet-scale distributed system, may be more vulnerable to different attacks aiming to jeopardize training or to infer some characteristics of the local dataset by looking at the different messages [26, 92]. Encryption [10, 80, 8] and differential privacy [1] techniques may help preventing such attacks.

Federated learning is less efficient than training in a highly-optimized computing cluster. It may in particular increase energy training costs, due to a more discontinuous usage of local computing resources and the additional cost of transmitting messages over long distance links. To the best of our knowledge, energetic considerations for federated learning have not been adequately explored, but for a few papers considering FL for mobile devices [42, 97].

# 7   Acknowledgements

The authors are grateful to the OPAL infrastructure from Université Côte d'Azur for providing computational resources and technical support.

This work was carried out and partially funded in the framework of a common lab agreement between Inria and Nokia Bell Labs (ADR 'Rethinking the Network').

The authors thank Damiano Carra, Alain Jean-Marie, Marco Lorenzi, and Pietro Michiardi for their feedback on early versions of this paper, François Baccelli, Bruno Gaujal, Laurent Hardouin, and Enrico Vicario for pointers to the literature of max-plus linear systems, and the Italian networking community (in particular Mauro Campanella, Marco Canini, Claudio Cicconetti, Francesca Cuomo, Paolo Giaccone, Dario Maggiorini, Marco Mellia, Antonio Pescapé, Tommaso Pecorella, and Luca Valcarenghi) for their suggestions to select realistic network scenarios for federated learning. Obviously, the authors keep the responsibility for any error in this paper.

## Footnotes

[1]The available bandwidth of a path is the maximum rate that the path can provide to a flow, taking into account the rest of the traffic [15, 39]; it is then smaller than the minimum link capacity of the path.

[2]A $\delta$-MBST is a spanning tree with degree at most $\delta$ in which the largest edge delay is as small as possible.

[3]The delay in the core network is determined by the available bandwidth as in (3). Available bandwidths are often limited to tens or hundreds of Mbps even over inter-datacenter links with capacities between 100 Gbps and 1 Tbps [38, 65, 83, 47]. By selecting 1 Gbps core links in our simulator, which ignores other traffic, we obtain available bandwidth distributions comparable to those observed in experimental studies like [38] (Appendix G).

[4]Additional experiments fine tuning $C_b$ were carried out, conclusions remain the same (Appendix H.6).

[5]Additional experiments were conducted selecting the matrix $A$ as solution of the fastest distributed linear averaging problem defined in [62] (Appendix H.4).

[6]As MATCHA and MATCHA$^{(+)}$ select random overlays at each iteration, we compute their average cycle time.

[7]Training time is evaluated as the time to reach a training accuracy equal to 65%, 55%, 55%, 50% and 50% for Gaia, AWS North America, Géant, Exodus, and Ebone networks, respectively. Note that data distribution is different in each network, so that a different global model is learned when solving Problem (1) (see explanations in Appendix H.5).

[8]In Appendix H.1, we show tables similar to Table 3 for different values of $s$.
