[Supplementary Material]



## Footnotes

[9] The set of subgraphs of an undirected graph $\mathcal{G}_c$ is finite.

[10]A decision problem is NP if we can verify in a polynomial time that the answer for a given instance is YES.

[11]See [73] for the proof.

[12]iNaturalist 2018 competition is part of the $FGVC^5$ workshop at CVPR (`https://github.com/visipedia/inat_comp/blob/master/2018/README.md`).

[13]The dataset size is reduced from 120GB to 18GB containing 67,000 images. We subsampled then $20\%$ from this dataset for training.

[14]Actually, the amount of data we considered is comparable to the federated learning paper [56]: we considered 10 times more data for FEMNIST and the same amount of data for Sentiment140 and Shakespeare.

[15]Training time is evaluated as the time to reach a top 5 training accuracy equal to $18\%$ for Gaia and to $13\%$ for other networks. The top 5 training accuracy reached by centralized training ResNet-50 after 50 epochs is about $20\%$.

# References

[1]     Martin Abadi et al. "Deep learning with differential privacy". In: *Proceedings of the 2016 ACM SIGSAC Conference on Computer and Communications Security*. 2016, pp. 308–318.

[2]     Patrick J. Andersen and Charl J. Ras. "Algorithms for Euclidean Degree Bounded Spanning Tree Problems". In: *Int. J. Comput. Geometry Appl.* 29.2 (2019), pp. 121–160.

[3]     Patrick J. Andersen and Charl J. Ras. "Minimum bottleneck spanning trees with degree bounds". In: *Networks* 68.4 (2016), pp. 302–314. DOI: 10.1002/net.21710.

[4]     David L. Applegate et al. *The Traveling Salesman Problem: A Computational Study (Princeton Series in Applied Mathematics)*. USA: Princeton University Press, 2007. ISBN: 0691129932.

[5]     Mahmoud Assran et al. "Stochastic Gradient Push for Distributed Deep Learning". In: *Proceedings of the 36th International Conference on Machine Learning, ICML 2019*. Vol. 97. Proceedings of Machine Learning Research. PMLR, 2019, pp. 344–353.

[6]     François Baccelli et al. "Synchronization and Linearity - An Algebra for Discrete Event Systems". In: *The Journal of the Operational Research Society* 45 (Jan. 1994). DOI: 10.2307/2583959.

[7]     Aurélien Bellet et al. "Personalized and Private Peer-to-Peer Machine Learning". In: *AISTATS*. 2018.

[8]     Keith Bonawitz et al. "Practical secure aggregation for privacy-preserving machine learning". In: *Proceedings of the 2017 ACM SIGSAC Conference on Computer and Communications Security*. 2017, pp. 1175–1191.

[9]     Keith Bonawitz et al. "Towards Federated Learning at Scale: System Design". In: *SysML 2019* abs/1902.01046 (2019).

[10]    Raphael Bost et al. "Machine learning classification over encrypted data." In: *NDSS*. Vol. 4324. 2015, p. 4325.

[11]    Ulrik Brandes. "On variants of shortest-path betweenness centrality and their generic computation". In: *Social Networks* 30.2 (2008), pp. 136–145. ISSN: 0378-8733. DOI: https://doi.org/10.1016/j.socnet.2007.11.001. URL: http://www.sciencedirect.com/science/article/pii/S0378873307000731.

[12]    T. Brunsch, J. Raisch, and L. Hardouin. "Modeling and control of high-throughput screening systems". In: *Control Engineering Practice* 20.1 (2012). Special Section: IFAC Conference on Analysis and Design of Hybrid Systems (ADHS'09) in Zaragoza, Spain, 16th-18th September, 2009, pp. 14–23. ISSN: 0967-0661. DOI: https://doi.org/10.1016/j.conengprac.2010.12.006. URL: http://www.sciencedirect.com/science/article/pii/S0967066110002662.

[13]    Sebastian Caldas et al. "Expanding the reach of federated learning by reducing client resource requirements". In: *arXiv preprint arXiv:1812.07210* (2018).

[14]    Sebastian Caldas et al. *LEAF: A Benchmark for Federated Settings*. 2018. arXiv: 1812.01097 [cs.LG].

[15]    Robert L. Carter and Mark E. Crovella. "Measuring bottleneck link speed in packet-switched networks". In: *Performance Evaluation* 27-28 (1996), pp. 297–318. ISSN: 0166-5316. DOI: https://doi.org/10.1016/S0166-5316(96)90032-2. URL: http://www.sciencedirect.com/science/article/pii/S0166531696900322.

[16]    Vigyan Chandra, Zhongdong Huang, and Ratnesh Kumar. "Automated control synthesis for an assembly line using discrete event system control theory". In: *IEEE Transactions on Systems, Man, and Cybernetics, Part C (Applications and Reviews)* 33.2 (2003), pp. 284–289.

[17]    Kyunghyun Cho et al. "On the Properties of Neural Machine Translation: Encoder-Decoder Approaches". In: *Proceedings of SSST@EMNLP 2014, Eighth Workshop on Syntax, Semantics and Structure in Statistical Translation, Doha, Qatar, 25 October 2014*. Ed. by Dekai Wu et al. Association for Computational Linguistics, 2014, pp. 103–111. DOI: 10.3115/v1/W14-4012. URL: https://www.aclweb.org/anthology/W14-4012/.

[18]    Igor Colin et al. "Gossip Dual Averaging for Decentralized Optimization of Pairwise Functions". In: *Proceedings of the 33rd International Conference on International Conference on Machine Learning - Volume 48*. ICML'16. New York, NY, USA: JMLR.org, 2016, pp. 1388–1396.

[19] Pierre Courtiol et al. "Deep learning-based classification of mesothelioma improves prediction of patient outcome". In: *Nature medicine* 25.10 (2019), pp. 1519–1525.

[20] A. Dasdan and R. K. Gupta. "Faster maximum and minimum mean cycle algorithms for system-performance analysis". In: *IEEE Transactions on Computer-Aided Design of Integrated Circuits and Systems* 17.10 (1998), pp. 889–899.

[21] X. David-Henriet et al. "Holding Time Maximization Preserving Output Performance for Timed Event Graphs". In: *IEEE Transactions on Automatic Control* 59.7 (2014), pp. 1968–1973.

[22] J. Deng et al. "ImageNet: A Large-Scale Hierarchical Image Database". In: *CVPR09*. 2009.

[23] P. Di Lorenzo and G. Scutari. "Distributed nonconvex optimization over time-varying networks". In: *2016 IEEE International Conference on Acoustics, Speech and Signal Processing (ICASSP)*. Mar. 2016, pp. 4124–4128. DOI: 10.1109/ICASSP.2016.7472453.

[24] J. C. Duchi, A. Agarwal, and M. J. Wainwright. "Dual Averaging for Distributed Optimization: Convergence Analysis and Network Scaling". In: *IEEE Transactions on Automatic Control* 57.3 (Mar. 2012), pp. 592–606. ISSN: 1558-2523. DOI: 10.1109/tac.2011.2161027. URL: http://dx.doi.org/10.1109/TAC.2011.2161027.

[25] N. Farhi, M. Goursat, and J.-P. Quadrat. "The traffic phases of road networks". In: *Transportation Research Part C: Emerging Technologies* 19.1 (2011), pp. 85–102. ISSN: 0968-090X. DOI: https://doi.org/10.1016/j.trc.2010.03.011. URL: http://www.sciencedirect.com/science/article/pii/S0968090X10000379.

[26] Matt Fredrikson, Somesh Jha, and Thomas Ristenpart. "Model inversion attacks that exploit confidence information and basic countermeasures". In: *Proceedings of the 22nd ACM SIGSAC Conference on Computer and Communications Security*. 2015, pp. 1322–1333.

[27] M. R. Garey and D. S. Johnson. *Computers and Intractability: A Guide to the Theory of NP-Completeness (Series of Books in the Mathematical Sciences)*. First Edition. W. H. Freeman, 1979. ISBN: 0716710455. URL: http://www.amazon.com/Computers-Intractability-NP-Completeness-Mathematical-Sciences/dp/0716710455.

[28] S. Gaubert. "Resource optimization and (min,+) spectral theory". In: *IEEE Transactions on Automatic Control* 40.11 (1995), pp. 1931–1934.

[29] *GÉANT - the pan-european research and education network*. URL: https://www.geant.org/Networks (visited on ).

[30] Alec Go, Richa Bhayani, and Lei Huang. "Twitter Sentiment Classification using Distant Supervision". In: *Processing* (2009), pp. 1–6. URL: http://www.stanford.edu/~alecmgo/papers/TwitterDistantSupervision09.pdf.

[31] Rob MP Goverde. "The max-plus algebra approach to railway timetable design". In: *WIT Transactions on The Built Environment* 37 (1998).

[32] Bamba Gueye et al. "Constraint-Based Geolocation of Internet Hosts". In: *Proceedings of the 4th ACM SIGCOMM Conference on Internet Measurement*. IMC '04. Taormina, Sicily, Italy: Association for Computing Machinery, 2004, pp. 288–293. ISBN: 1581138210. DOI: 10.1145/1028788.1028828. URL: https://doi.org/10.1145/1028788.1028828.

[33] Gregory Gutin and Abraham P Punnen. *The traveling salesman problem and its variations*. Vol. 12. Springer Science & Business Media, 2006.

[34] Aric A. Hagberg, Daniel A. Schult, and Pieter J. Swart. "Exploring Network Structure, Dynamics, and Function using NetworkX". In: *Proceedings of the 7th Python in Science Conference*. Ed. by Gaël Varoquaux, Travis Vaught, and Jarrod Millman. Pasadena, CA USA, 2008, pp. 11–15.

[35] Kaiming He et al. "Deep residual learning for image recognition". In: *Proceedings of the IEEE conference on computer vision and pattern recognition*. 2016, pp. 770–778.

[36] Michael Himsolt. *GML: A portable graph file format*. Tech. rep. Technical report, Universitat Passau, 1997.

[37] Sepp Hochreiter and Jürgen Schmidhuber. "Long Short-Term Memory". In: *Neural Computation* 9.8 (1997), pp. 1735–1780.

[38] Kevin Hsieh et al. "Gaia: Geo-Distributed Machine Learning Approaching LAN Speeds". In: *Proceedings of the 14th USENIX Conference on Networked Systems Design and Implementation*. NSDI'17. Boston, MA, USA: USENIX Association, 2017, pp. 629–647. ISBN: 9781931971379.

[39] Manish Jain and Constantinos Dovrolis. "End-to-End Available Bandwidth: Measurement Methodology, Dynamics, and Relation with TCP Throughput". In: *SIGCOMM Comput. Commun. Rev.* 32.4 (Aug. 2002), pp. 295–308. ISSN: 0146-4833. DOI: 10.1145/964725.633054. URL: https://doi.org/10.1145/964725.633054.

[40] Zhanhong Jiang et al. "Collaborative deep learning in fixed topology networks". In: *Advances in Neural Information Processing Systems*. 2017, pp. 5904–5914.

[41] Peter Kairouz et al. *Advances and Open Problems in Federated Learning*. 2019. arXiv: 1912.04977 [cs.LG].

[42] Jiawen Kang et al. "Incentive Design for Efficient Federated Learning in Mobile Networks: A Contract Theory Approach". In: *CoRR* abs/1905.07479 (2019). arXiv: 1905.07479. URL: http://arxiv.org/abs/1905.07479.

[43] Jerome J. Karaganis. "On the cube of a graph". In: 1968.

[44] Can Karakus et al. "Straggler Mitigation in Distributed Optimization Through Data Encoding". In: *Proc. of NIPS*. 2017, pp. 5434–5442.

[45] Sai Praneeth Karimireddy et al. *SCAFFOLD: Stochastic Controlled Averaging for Federated Learning*. 2019. arXiv: 1910.06378 [cs.LG].

[46] Richard M. Karp. "A characterization of the minimum cycle mean in a digraph". In: *Discrete Mathematics* 23.3 (1978), pp. 309–311. ISSN: 0012-365X. DOI: https://doi.org/10.1016/0012-365X(78)90011-0. URL: http://www.sciencedirect.com/science/article/pii/0012365X78900110.

[47] P. Kathiravelu et al. "Moving Bits with a Fleet of Shared Virtual Routers". In: *2018 IFIP Networking Conference (IFIP Networking) and Workshops*. 2018, pp. 1–9.

[48] S. Knight et al. "The Internet Topology Zoo". In: *Selected Areas in Communications, IEEE Journal on* 29.9 (Oct. 2011), pp. 1765–1775. ISSN: 0733-8716. DOI: 10.1109/JSAC.2011.111002.

[49] Anastasia Koloskova, Sebastian Stich, and Martin Jaggi. "Decentralized Stochastic Optimization and Gossip Algorithms with Compressed Communication". In: *Proceedings of the 36th International Conference on Machine Learning (ICML)*. Ed. by Kamalika Chaudhuri and Ruslan Salakhutdinov. Vol. 97. Proceedings of Machine Learning Research. Long Beach, California, USA: PMLR, June 2019, pp. 3478–3487. URL: http://proceedings.mlr.press/v97/koloskova19a.html.

[50] Anastasia Koloskova et al. *A Unified Theory of Decentralized SGD with Changing Topology and Local Updates*. 2020. arXiv: 2003.10422 [cs.LG].

[51] Jakub Konečný, Brendan McMahan, and Daniel Ramage. "Federated Optimization:Distributed Optimization Beyond the Datacenter". In: *8th NIPS Workshop on Optimization for Machine Learning (OPT15)*. 2015. arXiv: 1511.03575 [cs.LG].

[52] Jakub Konecný et al. "Federated Optimization: Distributed Machine Learning for On-Device Intelligence". In: *CoRR* abs/1610.02527 (2016). arXiv: 1610.02527. URL: http://arxiv.org/abs/1610.02527.

[53] Anusha Lalitha et al. "Peer-to-peer Federated Learning on Graphs". In: *CoRR* abs/1901.11173 (2019). arXiv: 1901.11173. URL: http://arxiv.org/abs/1901.11173.

[54] Jean-Yves Le Boudec and Patrick Thiran. *Network Calculus: A Theory of Deterministic Queuing Systems for the Internet*. Berlin, Heidelberg: Springer-Verlag, 2001. ISBN: 354042184X.

[55] Songze Li et al. "Near-Optimal Straggler Mitigation for Distributed Gradient Methods". In: *Proc. of the 7th Intl. Workshop ParLearning*. May 2018.

[56] Tian Li et al. "Federated Learning: Challenges, Methods, and Future Directions". In: *IEEE Signal Processing Magazine* 37.3 (2020), pp. 50–60.

[57] Tian Li et al. "Federated Optimization in Heterogeneous Networks". In: *Proceedings of the 3rd MLSys Conference* (2020).

[58] Xiang Li et al. "Communication-Efficient Local Decentralized SGD Methods." In: *arXiv: Machine Learning* (2019).

[59] Athanassios Liakopoulos et al. "Providing and verifying advanced IP services in hierarchical DiffServ networks-the case of GEANT". In: *International Journal of Communication Systems* 17.4 (2004), pp. 321–336. DOI: 10.1002/dac.645. eprint: https://onlinelibrary.wiley.com/doi/pdf/10.1002/dac.645.

[60] Xiangru Lian et al. "Asynchronous Decentralized Parallel Stochastic Gradient Descent". In: *Proceedings of the 35th International Conference on Machine Learning*. Ed. by Jennifer Dy and Andreas Krause. Vol. 80. Proceedings of Machine Learning Research. Stockholmsmässan, Stockholm Sweden: PMLR, July 2018, pp. 3043–3052.

[61] Xiangru Lian et al. "Can Decentralized Algorithms Outperform Centralized Algorithms? A Case Study for Decentralized Parallel Stochastic Gradient Descent". In: *Advances in Neural Information Processing Systems 30*. Ed. by I. Guyon et al. Curran Associates, Inc., 2017, pp. 5330–5340.

[62] Lin Xiao and S. Boyd. "Fast linear iterations for distributed averaging". In: *42nd IEEE International Conference on Decision and Control (IEEE Cat. No.03CH37475)*. Vol. 5. Dec. 2003, 4997–5002 Vol.5. DOI: 10.1109/CDC.2003.1272421.

[63] J. Lin. "Divergence measures based on the Shannon entropy". In: *IEEE Transactions on Information Theory* 37.1 (1991), pp. 145–151.

[64] Tao Lin et al. "Don't Use Large Mini-batches, Use Local SGD". In: *International Conference on Learning Representations*. 2020. URL: https://openreview.net/forum?id=B1eyO1BFPr.

[65] S. Liu and B. Li. "Stemflow: Software-Defined Inter-Datacenter Overlay as a Service". In: *IEEE Journal on Selected Areas in Communications* 35.11 (2017), pp. 2563–2573.

[66] Qinyi Luo et al. "Hop: Heterogeneity-Aware Decentralized Training". In: *Proceedings of the Twenty-Fourth International Conference on Architectural Support for Programming Languages and Operating Systems - ASPLOS '19* (2019), pp. 893–907. DOI: 10.1145/3297858.3304009. URL: http://dx.doi.org/10.1145/3297858.3304009.

[67] *Machine learning ledger orchestration for drug discovery (MELLODY)*. EU research project. 2019. URL: https://www.imi.europa.eu/projects-results/project-factsheets/melloddy.

[68] Ratul Mahajan et al. "Inferring Link Weights using End-to-End Measurements". In: *Workshop on Internet measurment (IMW)*. Aug. 2002.

[69] *Mammogram Assessment with NVIDIA Clara Federated Learning*. EU research project. 2020. URL: https://blogs.nvidia.com/blog/2020/04/15/federated-learning-mammogram-assessment/.

[70] Sébastien Marcel and Yann Rodriguez. "Torchvision the Machine-Vision Package of Torch". In: *Proceedings of the 18th ACM International Conference on Multimedia*. MM '10. Firenze, Italy: Association for Computing Machinery, 2010, pp. 1485–1488. ISBN: 9781605589336. DOI: 10.1145/1873951.1874254. URL: https://doi.org/10.1145/1873951.1874254.

[71] L. Massoulie et al. "Randomized Decentralized Broadcasting Algorithms". In: *Proceedings of the IEEE INFOCOM 2007 - 26th IEEE International Conference on Computer Communications*. USA: IEEE Computer Society, 2007, pp. 1073–1081. ISBN: 1424410479. DOI: 10.1109/INFCOM.2007.129. URL: https://doi.org/10.1109/INFCOM.2007.129.

[72] Brendan McMahan et al. "Communication-Efficient Learning of Deep Networks from Decentralized Data". In: *Proceedings of the 20th International Conference on Artificial Intelligence and Statistics, AISTATS 2017,* ed. by Aarti Singh and Xiaojin (Jerry) Zhu. Vol. 54. Proceedings of Machine Learning Research. PMLR, 2017, pp. 1273–1282.

[73] Jérôme Monnot, Vangelis Th. Paschos, and Sophie Toulouse. "Approximation algorithms for the traveling salesman problem". In: *Mathematical Models of Operations Research* 56 (2002), pp. 387–405. URL: https://hal.archives-ouvertes.fr/hal-00003997.

[74] *Musketeer*. URL: http://musketeer.eu/project/.

[75] A. Nedić, A. Olshevsky, and M. G. Rabbat. "Network Topology and Communication-Computation Tradeoffs in Decentralized Optimization". In: *Proceedings of the IEEE* 106.5 (May 2018), pp. 953–976. ISSN: 1558-2256. DOI: 10.1109/JPROC.2018.2817461.

[76] Angelia Nedic, Alex Olshevsky, and Wei Shi. "Achieving Geometric Convergence for Distributed Optimization Over Time-Varying Graphs". In: *SIAM J. Optimization* 27.4 (2017), pp. 2597–2633.

[77] Angelia Nedić and Asuman E. Ozdaglar. "Distributed Subgradient Methods for Multi-Agent Optimization". In: *IEEE Trans. Automat. Contr.* 54.1 (2009), pp. 48–61.

[78] G. Neglia et al. "The Role of Network Topology for Distributed Machine Learning". In: *IEEE INFOCOM 2019 - IEEE Conference on Computer Communications*. Apr. 2019, pp. 2350–2358. DOI: 10.1109/INFOCOM.2019.8737602.

[79] Giovanni Neglia et al. "Decentralized gradient methods: does topology matter?" In: *AISTATS 2020 - 23rd International Conference on Artificial Intelligence and Statistics*. Palermo, Italy, June 2020. URL: https://hal.inria.fr/hal-02430485.

[80] Valeria Nikolaenko et al. "Privacy-preserving ridge regression on hundreds of millions of records". In: *2013 IEEE Symposium on Security and Privacy*. IEEE. 2013, pp. 334–348.

[81] Alex Olshevsky, Ioannis Ch. Paschalidis, and Shi Pu. *Asymptotic Network Independence in Distributed Optimization for Machine Learning*. 2019. arXiv: 1906.12345 [math.OC].

[82] Jeffrey Pennington, Richard Socher, and Christopher D Manning. "Glove: Global Vectors for Word Representation." In: *EMNLP*. Vol. 14. 2014, pp. 1532–1543.

[83] Valerio Persico et al. "On the performance of the wide-area networks interconnecting public-cloud datacenters around the globe". In: *Computer Networks* 112 (2017), pp. 67–83. ISSN: 1389-1286. DOI: https://doi.org/10.1016/j.comnet.2016.10.013. URL: http://www.sciencedirect.com/science/article/pii/S138912861630353X.

[84] R. Prasad et al. "Bandwidth estimation: metrics, measurement techniques, and tools". In: *IEEE Network* 17.6 (2003), pp. 27–35.

[85] R. C. Prim. "Shortest Connection Networks And Some Generalizations". In: *Bell System Technical Journal* 36.6 (1957), pp. 1389–1401. DOI: 10.1002/j.1538-7305.1957.tb01515.x.

[86] Shi Pu, Alex Olshevsky, and Ioannis Ch. Paschalidis. "Asymptotic Network Independence in Distributed Stochastic Optimization for Machine Learning: Examining Distributed and Centralized Stochastic Gradient Descent". In: *IEEE Signal Process. Mag.* 37.3 (2020), pp. 114–122.

[87] S. Sundhar Ram, Angelia Nedic, and Venugopal V. Veeravalli. "A new class of distributed optimization algorithms: application to regression of distributed data". In: *Optimization Methods and Software* 27.1 (2012), pp. 71–88. DOI: 10.1080/10556788.2010.511669. URL: https://doi.org/10.1080/10556788.2010.511669.

[88] Felix Sattler et al. "Robust and communication-efficient federated learning from non-iid data". In: *IEEE transactions on neural networks and learning systems* (2019).

[89] Kevin Scaman et al. "Optimal algorithms for non-smooth distributed optimization in networks". In: *Advances in Neural Information Processing Systems*. 2018, pp. 2740–2749.

[90] Kevin Seaman et al. "Optimal algorithms for smooth and strongly convex distributed optimization in networks". In: *Proceedings of the 34th International Conference on Machine Learning-Volume 70*. JMLR. org. 2017, pp. 3027–3036.

[91] Wei Shi et al. "EXTRA: An Exact First-Order Algorithm for Decentralized Consensus Optimization". In: *SIAM J. Optimization* 25.2 (2015), pp. 944–966.

[92] Reza Shokri et al. "Membership inference attacks against machine learning models". In: *2017 IEEE Symposium on Security and Privacy (SP)*. IEEE. 2017, pp. 3–18.

[93] Santiago Silva et al. "Federated learning in distributed medical databases: Meta-analysis of large-scale subcortical brain data". In: *2019 IEEE 16th International Symposium on Biomedical Imaging (ISBI 2019)*. IEEE. 2019, pp. 270–274.

[94] Neil Spring et al. "Measuring ISP Topologies with Rocketfuel". In: *IEEE/ACM Trans. Netw.* 12.1 (Feb. 2004), pp. 2–16. ISSN: 1063-6692. DOI: 10.1109/TNET.2003.822655. URL: https://doi.org/10.1109/TNET.2003.822655.

[95] Hanlin Tang et al. "$D^2$: Decentralized Training over Decentralized Data". In: *Proceedings of the 35th International Conference on Machine Learning*. Ed. by Jennifer Dy and Andreas Krause. Vol. 80. Proceedings of Machine Learning Research. Stockholmsmässan, Stockholm Sweden: PMLR, July 2018, pp. 4848–4856. URL: http://proceedings.mlr.press/v80/tang18a.html.

[96] *The AWS Cloud in North America*. URL: https://aws.amazon.com/about-aws/global-infrastructure/?nc1=h_ls.

[97] N. H. Tran et al. "Federated Learning over Wireless Networks: Optimization Model Design and Analysis". In: *IEEE INFOCOM 2019 - IEEE Conference on Computer Communications*. 2019, pp. 1387–1395.

[98]  K. I. Tsianos, S. Lawlor, and M. G. Rabbat. "Consensus-based distributed optimization: Practical issues and applications in large-scale machine learning". In: *2012 50th Annual Allerton Conference on Communication, Control, and Computing (Allerton)*. Oct. 2012, pp. 1543–1550. DOI: 10.1109/Allerton.2012.6483403.

[99]  G. Van Horn et al. "The iNaturalist Species Classification and Detection Dataset". In: *2018 IEEE/CVF Conference on Computer Vision and Pattern Recognition*. 2018, pp. 8769–8778.

[100] Paul Vanhaesebrouck, Aurélien Bellet, and Marc Tommasi. "Decentralized Collaborative Learning of Personalized Models over Networks". In: *AISTATS*. 2017.

[101] Hongyi Wang et al. "Atomo: Communication-efficient learning via atomic sparsification". In: *Advances in Neural Information Processing Systems*. 2018, pp. 9850–9861.

[102] Jianyu Wang and Gauri Joshi. "Adaptive communication strategies to achieve the best error-runtime trade-off in local-update SGD". In: *MLSys* (2019).

[103] Jianyu Wang and Gauri Joshi. "Cooperative SGD: A unified Framework for the Design and Analysis of Communication-Efficient SGD Algorithms". In: *ICML Workshop* (2019).

[104] Jianyu Wang et al. "MATCHA: Speeding Up Decentralized SGD via Matching Decomposition Sampling". In: *NIPS Workshop* (2019).

[105] Jianyu Wang et al. *SlowMo: Improving Communication-Efficient Distributed SGD with Slow Momentum*. 2019. arXiv: 1910.00643 [cs.LG].

[106] Shiqiang Wang et al. "Adaptive Federated Learning in Resource Constrained Edge Computing Systems". In: *IEEE J. Sel. Areas Commun.* 37.6 (2019), pp. 1205–1221. DOI: 10.1109/JSAC.2019.2904348. URL: https://doi.org/10.1109/JSAC.2019.2904348.

[107] *WeBank*. URL: https://finance.yahoo.com/news/webank-swiss-signed-cooperation-mou-112300218.html;https://fate.fedai.org/.

[108] Blake Woodworth et al. *Is Local SGD Better than Minibatch SGD?* 2020. arXiv: 2002.07839 [cs.LG].

[109] Jinshan Zeng Yin and Wotao. "Extrapush for Convex Smooth Decentralized Optimization Over Directed Networks". In: *Journal of Computational Mathematics* 35.4 (June 2017), pp. 383–396. ISSN: 1991-7139.

[110] K. Yuan et al. "Exact Diffusion for Distributed Optimization and Learning—Part I: Algorithm Development". In: *IEEE Transactions on Signal Processing* 67.3 (2019), pp. 708–723.

[111] Hantian Zhang et al. "ZipML: Training Linear Models with End-to-End Low Precision, and a Little Bit of Deep Learning". In: *Proceedings of the 34th International Conference on Machine Learning*. Vol. 70. Proceedings of Machine Learning Research. PMLR, 2017, pp. 4035–4043.

# A Graph Theory

We now list concepts of graph theory which will be used later on.

- **Predecessor, successor, neighbour**: If in a graph $(i, j) \in \mathcal{E}$, then $i$ is called a predecessor of $j$, $j$ is called a successor of $i$ and $j$, resp. $i$ is called a neighbour of $i$, resp. $j$. The set of predecessors of $j$ is indicated by $\pi(j)$ (or $\mathcal{N}_j^+$), the set of all successors of $i$ is denoted $\sigma(i)$ (or $\mathcal{N}_i^-$) and the set of neighbours of $i$ is denoted $\mathcal{N}_i$. Note that in the case of undirected graphs, $\mathcal{N}_i = \pi(i) = \sigma(i)$.

- **Path, circuit**: A path is a sequence of nodes $(i_1, \ldots, i_p), p > 1$, such that $i_j \in \pi(i_{j+1}), j = 1, \ldots, p-1$. An elementary path is a path where no node appears more then once. When the initial node and the final node coincide, we call the path a circuit. A circuit $C = (i_1, \ldots, i_p = i_1)$ is an elementary circuit if the path $(i_1, \ldots, i_{p-1})$ is elementary, an elementary circuit is sometimes referred to as a cycle. If a cycle spans all vertices of the graph it is called a *Hamiltonian cycle*. The length of circuit $C = (i_1, \ldots, i_p)$ is the number of the arcs of which it is composed, i.e., $|C| = p$, and its weight is the sum of the weights of its arcs, i.e, $d(C) = \sum_{k=1}^{p-1} d(i_k, i_{k+1})$.

- **Subgraph, spanning subgraph**: Given a graph $\mathcal{G} = (\mathcal{V}, \mathcal{E})$, a graph $\mathcal{G}' = (\mathcal{V}', \mathcal{E}')$ is said to be a subgraph of $\mathcal{G}$ if $\mathcal{V}' \subset \mathcal{V}$ and $\mathcal{E} \subset \mathcal{E}'$. $\mathcal{G}'$ is said to be a spanning subgraph if $\mathcal{V}' = \mathcal{V}$.

- **Strongly connected graph**: A digraph is said to be *strongly connected* or *strong* if for any two different nodes $i$ and $j$ in $\mathcal{V}$ there exists a path from $i$ to $j$.

- **Optimal tour**: In a Hamiltonian graph (i.e., a graph having a Hamiltonian cycle) a Hamiltonian cycle with minimum weight is called an *optimal tour*. Finding the optimal tour in a complete graph is a well known problem and is referred to as the Traveling Salesman Problem (TSP), see for example [4].

- **Tree, acyclic graph, and Minimum Spanning Tree (MST)**: A tree, or equivalently a connected acyclic undirected graph, is an undirected graph in which any two vertices are connected by exactly one path. An acyclic graph $\mathcal{T}$ is said to be a spanning tree of an undirected graph $\mathcal{G}$ if $\mathcal{T}$ is a connected spanning subgraph of $\mathcal{G}$. $\mathcal{T}$ is said to be an MST of $\mathcal{G}$ if it has minimal weight (the weight of a tree is the sum of the weights of all its edges) among all spanning trees of $\mathcal{G}$.

- **Cut, cut-set, and cut property**: A *cut* is a partition of the vertices of a graph into two disjoint subsets. For a cut $c$, the cut-set is the set of edges connecting two nodes from the two disjoint subsets. In a tree, deleting an edge, induces a partition of the set of vertices. For any cut $c$ of the graph, if the weight of an edge $e$ in the cut-set of $c$ is strictly smaller than the weights of all other edges of the cut-set of $c$, then this edge belongs to all MSTs of the graph.

# B  On STAR and MATCHA$^{(+)}$ Cycle Times

For a graph $\mathcal{G}$, let $\mathrm{degree}(i, \mathcal{G})$ denote the degree node $i$ in $\mathcal{G}$ and $\max(\mathrm{degree}(\mathcal{G}))$ denote the maximum degree of the nodes in $\mathcal{G}$. We show that, with $N$ silos, the RING is up to $2N$ times faster than the STAR and approximately $C_b \times \max(\mathrm{degree}(\mathcal{G}_u))$ times faster then MATCHA$^{(+)}$ for slow homogeneous access links as shown also in Fig. 3a.

Since access links are homogeneous, i.e., $C_{\mathrm{UP}}(i) = C_{\mathrm{DN}}(i) = C_{\mathrm{UP}}(j) = C_{\mathrm{DN}}(j) = C, \forall i, j \in \mathcal{V}$, and slow access links determine the delays, i.e., $C_{\mathrm{UP}}(i) \ll A(i', j')$ and $s \times T_c(i) + l(i, j) \ll \frac{M}{A(i,j)}$, according to (3), we have:

$$d_o(i, j) = \max\left(|\mathcal{N}_i^-|, |\mathcal{N}_j^+|\right) \times \frac{M}{C}.$$

Then, the cycle time of the RING can be obtained from (5):

$$\tau_{\mathrm{RING}} = \frac{\sum_{i=1}^{N} d_o(i, i+1)}{N} = \frac{\frac{M}{C} \times N}{N} = \frac{M}{C}.$$

Remember that a cycle is the time interval between two consecutive computations at a given silo. For the STAR, it corresponds to the time interval between when the central node sends the new aggregate model to all silos and when it receives all updated local models. Therefore, we have:

$$\tau_{\mathrm{STAR}} = \frac{M}{C} \times N + \frac{M}{C} \times N = 2N \times \frac{M}{C}.$$

For MATCHA$^+$, at each communication round, we select a random subgraph $\mathcal{G}$. Let $\mathrm{degree}(i, \mathcal{G})$ denote the degree of silo $i$ in $\mathcal{G}$. If $\mathcal{G}$ is drawn, the duration of the communication round is $M/C \times \max(\mathrm{degree}(\mathcal{G}))$. The cycle time is then

$$\tau_{\mathrm{MATCHA}^+} = \frac{M}{C} \mathbb{E}_\mathcal{G}\left[\max \mathrm{degree}(\mathcal{G})\right].$$

Let $j$ be the silo such that $j'$ has the largest degree in $\mathcal{G}_u$. MATCHA$^+$ uses $\max(\mathrm{degree}(\mathcal{G}_u)) + 1$ matchings. The edges of $j$ belong to different matchings. As MATCHA$^+$ activates at any communication round a fraction $C_b$ of all matchings, the average degree of node $j$ is $\mathbb{E}_\mathcal{G}\left[\mathrm{degree}(j, \mathcal{G})\right] \approx C_b \times \mathrm{degree}(j, \mathcal{G}_u) = C_b \times \max(\mathrm{degree}(\mathcal{G}_u))$. Then

$$\tau_{\mathrm{MATCHA}^+} \gtrapprox \frac{M}{C} \times C_b \times \max(\mathrm{degree}(\mathcal{G}_u)).$$

(a) A 3-node example.　　　　　　　　(b) Example with arbitrarily different cycle times.

Figure 5: Networks where a directed topology outperforms an undirected one.

## C  Directed Overlays may be Faster than Undirected Overlays

We provide two examples where the underlay network is undirected and still a directed overlay can have shorter cycle time than directed overlays. Examples are in Fig. 5, where numbers associated to links are the corresponding delays (in the two directions).

The network in Fig. 5a has only three nodes, $\mathcal{V} = \{1, 2, 3\}$. We have $d_c(1, 2) = d_c(2, 1) = 1$, $d_c(2, 3) = d_c(3, 2) = 3$, and $d_c(1, 3) = d_c(3, 1) = 4$. The fastest undirected overlay is $G_o^{(u)} = (\mathcal{V}, \{(1, 2), (2, 3)\})$. Consider the directed ring $G_o = (\mathcal{V}, \{(1, 2), (2, 3), (3, 1)\})$. We have:

$$\tau\left(\mathcal{G}_o^{(u)}\right) = \max\left(\frac{1+1}{2}, \frac{3+3}{2}, \frac{1+3+1+3}{4}\right) = 3, \tag{7}$$

$$\tau\left(\mathcal{G}_o\right) = \frac{1 + 3 + (3+1)}{3} = \frac{8}{3} < 3. \tag{8}$$

The network in Fig. 5b shows that a directed ring can be arbitrarily faster than an undirected one. Similarly to above, the fastest undirected overlay is $G_o^{(u)}$ and coincides with the underlay. The directed overlay is the ring $(1 \rightarrow 2 \rightarrow 3 \rightarrow \ldots n \rightarrow n + 1 \rightarrow 1)$. We have

$$\tau\left(\mathcal{G}_o^{(u)}\right) = n, \tag{9}$$

$$\tau\left(\mathcal{G}_o\right) = \frac{(n-1) \times 1 + n + (n + (n-1) \times 1)}{n+1} = \frac{4n-2}{n+1} < 4. \tag{10}$$

The ratio of the two cycle times can be made arbitrarily large.

# D Approximation Algorithm for MCT on Node-Capacitated Networks

In this section, we describe Algorithm 1 that provides an approximate solution for MCT when the network is node-capacitated and $\mathcal{G}_c$ is complete. Algorithm 1 combines existing approximation algorithms for $\delta$-MBST on a particular undirected graph built from $\mathcal{G}_c$ and denoted by $\mathcal{G}_c^{(u)}$ (lines 1-3). Lemma E.5 establishes a connection between the bottleneck of the MBST of $\mathcal{G}_c^{(u)}$ and the cycle time of MCT on $\mathcal{G}_c$ when the overlay is required to be undirected. To get an approximated 2-MBST on $\mathcal{G}_c^{(u)}$, we apply the best known 3-approximation algorithm from [3, Sect. 3.2.1] (lines 6-8) which requires $\mathcal{G}_c^{(u)}$ to be Euclidean (Lemma E.6), and take its result as one candidate for our solution (line 9). The cube of a graph $\mathcal{G}$, denoted by $\mathcal{G}^3$, is the super-graph of $\mathcal{G}$ such that the edge $(u, v)$ is in $\mathcal{G}^3$ if and only if there is a path between $u$ and $v$ in $\mathcal{G}$ with three or fewer edges. It has been proved that the cube of a connected graph is Hamiltonian and to find a Hamiltonian path in such a cube can be done in polynomial time [43]. Other $\delta$-BSTs built by Algorithm 2 for $3 \leq \delta \leq N$ are considered as candidates (lines 10-11) and we finally provide as solution the overlay with the smallest cycle time (line 13).

---

**Algorithm 1:** Approximation algorithm for MCT on node-capacitated networks.

---

**Input :** $\mathcal{G}_c = (\mathcal{V}, \mathcal{E}_c)$, uplink capacity $C_{\text{UP}}(i)$, end-to-end delay $l(i, j)$, computation time $T_c(i)$ and model size $M$.

**Result:** Undirected overlay $\mathcal{G}_o$.

1 Create $\mathcal{G}_c^{(u)} = (\mathcal{V}, \mathcal{E}_c^{(u)})$ where $(i, j) \in \mathcal{E}_c^{(u)}$ iff $(i, j) \in \mathcal{E}_c$ and $(j, i) \in \mathcal{E}_c$ ;

2 **for** $(i, j) \in \mathcal{E}_c^{(u)}$ **do**

3 $\quad$ $d_c^{(u)}(i, j) = [s \times (T_c(i) + T_c(j)) + l(i, j) + l(j, i) + \frac{M}{C_{\text{UP}}(i)} + \frac{M}{C_{\text{UP}}(j)}]/2$

4 **end**

5 $\mathbb{S} \leftarrow \emptyset$ ; `// the set of candidate solutions`

$\quad$ `/* consider 2-MBST approximate solution on` $\mathcal{G}_c^{(u)}$ `as one candidate    */`

6 $\mathcal{T} \leftarrow$ a minimum weight spanning tree of $\mathcal{G}_c^{(u)}$ ;

7 $\mathcal{T}^3 \leftarrow$ the cube of $\mathcal{T}$ ;

8 $\mathcal{H} \leftarrow$ a Hamiltonian path in $\mathcal{T}^3$ ;

9 $\mathbb{S} \leftarrow \{\mathcal{H}\}$;

$\quad$ `/* consider other` $\delta$`-BST for` $3 \leq \delta \leq N$ `as candidates                  */`

10 **for** $\delta \in \{3, 4, 5, ..., N\}$ **do**

11 $\quad$ $\mathbb{S} \leftarrow \mathbb{S} \cup \{\delta\text{-PRIM}(\mathcal{G}_c^{(u)})\}$ `//` $\delta$`-PRIM(`$\mathcal{G}_c^{(u)}$`) gives a` $\delta$`-BST on` $\mathcal{G}_c^{(u)}$

12 **end**

$\quad$ `/* choose the one with the minimum cycle time as output overlay    */`

13 $\mathcal{G}_o \leftarrow \arg\min_{G \in \mathbb{S}} \tilde{\tau}(G)$

---

**Algorithm 2:** $\delta$-PRIM[2]

---

1 **Function** $\delta$-PRIM$(\mathcal{G} = (\mathcal{V}, \mathcal{E}))$**:**

2 $\quad$ $\mathcal{V}_T := \{v_0\}$ for some $v_0 \in \mathcal{V}$;

3 $\quad$ $\mathcal{E}_T := \{\}$;

4 $\quad$ $T = (\mathcal{V}_T, \mathcal{E}_T)$;

5 $\quad$ **while** $|\mathcal{E}_T| < |\mathcal{V}| - 1$ **do**

6 $\quad\quad$ Find the smallest weight edge $(u, v)$ such that $u \in \mathcal{V}_T$, $v \notin \mathcal{V}_T$, and $\text{DEGREE}_T(u) < \delta$;

7 $\quad\quad$ Add $v$ to $\mathcal{V}_T$;

8 $\quad\quad$ Add $(u, v)$ to $\mathcal{E}_T$;

9 $\quad$ **end**

10 $\quad$ **return** $T$ ;

---

# E  Proofs

We use some graph terminology and notation introduced in Appendix A.

## E.1  Proof of Proposition 3.1

When we require the overlay $\mathcal{G}_o$ to be undirected, if we include link $(i,j) \in \mathcal{G}_c$ then we will also include link $(j,i)$. It is then convenient to consider the undirected graph $\mathcal{G}_c^{(u)} = (\mathcal{V}, \mathcal{E}_c^{(u)})$, where $(i,j) \in \mathcal{E}_c^{(u)}$ iff $(i,j) \in \mathcal{E}_c$ and $(j,i) \in \mathcal{E}_c$, from which we want to extract an undirected strong subgraph $\mathcal{G}_o$ with minimal cycle time. We also associate to each edge $(i,j) \in \mathcal{G}_c^{(u)}$ the weight $d_c^{(u)}(i,j) = (d_c(i,j) + d_c(j,i))/2$. Remember that $d_c(i,j)$ is defined as follows

$$d_c(i,j) \triangleq s \times T_c(i) + l(i,j) + M/A(i',j').$$

Note that an undirected weighted graph can be also seen as a particular directed graph where for each link $(i,j)$ in one direction, there exists a link $(j,i)$ with the opposite direction and the same weight. The concept of cycle time can then immediately be extended to undirected graphs.

**Lemma E.1.** *Consider the undirected weighted graph $\mathcal{G}_c^{(u)} = (\mathcal{V}, \mathcal{E}_c^{(u)})$, where $(i,j) \in \mathcal{E}_c^{(u)}$ iff $(i,j) \in \mathcal{E}_c$ and $(j,i) \in \mathcal{E}_c$. When $\mathcal{G}_c$ is edge-capacitated and $\mathcal{G}_o$ is required to be undirected, the set of solutions* MCT *includes a spanning tree of $\mathcal{G}_c^{(u)}$.*

*Proof.* MCT is a discrete optimization problem on a finite set,[9] thus the set of solutions of MCT is non-empty. Suppose by contradiction that the set of solutions does not contain any spanning tree of $\mathcal{G}_c$ and consider $\mathcal{G}_o^*$ to be one of such solutions.

As $\mathcal{G}_o^*$ is not a spanning tree and it is strongly connected, there exist circuits in $\mathcal{G}_o^*$. For any circuit $C = (i_1, i_2, \ldots, i_p = i_1)$ in $\mathcal{G}_o^*$, we consider the edge $e_C$, such that $d_c^{(u)}(e_C) = \max_{k=1,\ldots,p-1} d_c^{(u)}(i_k, i_{k+1})$. The graph $\hat{\mathcal{G}}_o^*$ obtained from $\mathcal{G}_o^*$ by deleting $e_C$ is a connected spanning subgraph of $\mathcal{G}_c^{(u)}$ and its cycle time is not greater then the cycle time of $\mathcal{G}_o^*$. We can now proceed in the same way on $\hat{\mathcal{G}}_o^*$ until the residual graph has no more circuits and it is then a spanning tree of $\mathcal{G}_c^{(u)}$ with cycle time not greater than the cycle time of $\mathcal{G}_o^*$. This tree is also a solution of MCT contradicting the fact that no spanning tree is in the set of solutions. $\square$

**Lemma E.2.** *Consider an undirected tree $\mathcal{T} = (\mathcal{V}, \mathcal{E})$, weighted with a delay function $d_c^{(u)} : \mathcal{V} \times \mathcal{V} \mapsto \mathbb{R}_+$. Its cycle time is $\tau(\mathcal{T}) = \max_{\{i,j\} \in \mathcal{E}} d_c^{(u)}(i,j)$.*

*Proof.* The cycle time of $\mathcal{T}$ is given by Equation (5). $\tau(\mathcal{T}) = \max_C \frac{w(C)}{|C|}$, where the maximum is taken over all the elementary circuits of $\mathcal{T}$. Since $\mathcal{T}$ is acyclic, the only elementary circuits of $\mathcal{T}$ are of the form $(i,j,i)$ for some $\{i,j\} \in \mathcal{E}$. By definition $|(i,j,i)| = 2$ and $w((i,j,i)) = d_c^{(u)}(i,j)$. It follows that $\tau(\mathcal{T}) = \max_{\{i,j\} \in \mathcal{E}} \frac{d_c^{(u)}(i,j) + d_c^{(u)}(j,i)}{2} = \max_{\{i,j\} \in \mathcal{E}} d_c^{(u)}(i,j)$. $\square$

**Proposition 3.1.** *Consider an undirected weighted graph $\mathcal{G}_c^{(u)} = (\mathcal{V}, \mathcal{E}_c^{(u)})$, where $(i,j) \in \mathcal{E}_c^{(u)}$ iff $(i,j) \in \mathcal{E}_c$ and $(j,i) \in \mathcal{E}_c$ and where $(i,j) \in \mathcal{E}_c^{(u)}$ has weight $d_c^{(u)}(i,j) = (d_o(i,j) + d_o(j,i))/2$. A minimum weight spanning tree of $\mathcal{G}_c^{(u)}$ is a solution of* MCT *when $\mathcal{G}_c$ is edge-capacitated and $\mathcal{G}_o$ is required to be undirected.*

*Proof.* Denote by $\mathcal{G}^*$ the solution of MCT when $\mathcal{G}_c$ is edge-capacitated and $\mathcal{G}_o$ is required to be undirected, and denote $\mathcal{T}^*$ an MST of $\mathcal{G}_c^{(u)}$ weighted with $d_c^{(u)}$, and suppose by contradiction that $\tau(\mathcal{T}^*) > \tau(\mathcal{G}^*)$. By Lemma E.2, it follows that there is an edge $e_{\mathcal{T}^*}$ of $\mathcal{T}^*$ such that $d_c^{(u)}(e_{\mathcal{T}^*}) = \tau(\mathcal{T}^*)$. Moreover, it follows that $\forall e \in \mathcal{E}(\mathcal{G}^*)$, $d_c^{(u)}(e) \leq \tau(\mathcal{G}^*) < \tau(\mathcal{T}^*) = d_c^{(u)}(e_{\mathcal{T}^*})$. If we remove $e_{\mathcal{T}^*}$ from $\mathcal{T}^*$, the two components define a cut of $\mathcal{G}_c$. The edge of $\mathcal{G}^*$, say $e_{cut}$ belonging to the cut-set is such that $d_c^{(u)}(e_{cut}) < d_c^{(u)}(e_{\mathcal{T}^*})$, and this is a contradiction with the cut property satisfied by minimum cost spanning trees. $\square$

## E.2 Proof of Proposition 3.2

**Proposition 3.2.** MCT *is NP-hard even when* $\mathcal{G}_c$ *is a complete Euclidean edge-capacitated graph.*

*Proof.* When $\mathcal{G}_c$ is an edge-capacitated graph, $d_c(i,j) = s \times T_c(i) + l(i,j) + \frac{M}{A(i',j')}$. $\mathcal{G}_c$ is complete and Euclidean means that $d_c(i,j) = d_c(j,i)$, for all $(i,j) \in \mathcal{V} \times \mathcal{V}$ and that $d_c$ verifies triangular inequality, i.e., $d_c(i,j) \leq d_c(i,k) + d_c(k,j)$, for every $i,j,k \in \mathcal{V}$.

We consider the decision problem MCT-DECISION associated to the particular case of MCT when $\mathcal{G}_c$ is an Euclidean edge-capacitated graph and we prove that it is NP-complete.

Euclidean Edge-Capacitated Minimal Cycle Time - Decision (MCT-DECISION)
**Input:** A strong digraph $\mathcal{G}_c = (\mathcal{V}, \mathcal{E}_c)$, delays function $d_c$ and a real number $\tau_0$
**Output:** Is there a strong spanning subdigraph of $\mathcal{G}_c$ with cycle time at most $\tau_0$?

We first prove that MCT-DECISION is NP.[10] Several algorithms (e.g., Karp's Algorithm [20]) determines the cycle time of a given graph in a polynomial time. Thus for a proposed solution of MCT-DECISION, we can compute its cycle time in polynomial time, and we can verify if the graph is strongly connected using for example depth first search. It follows that MCT-DECISION is NP.

To prove that MCT-DECISION is NP-complete, we show that Hamiltonian Cycle (HC) can be reduced in a polynomial time to MCT-DECISION, i.e., HC $\leq_p$ MCT-DECISION.

Hamiltonian cycle problem is the following decision problem:

Hamiltonian Cycle (HC)
**Input:** A connected (undirected) graph $\mathcal{D} = (\mathcal{V}, \mathcal{E})$.
**Output:** Is there a Hamiltonian cycle in $\mathcal{D}$?

Given an instance of HC with an undirected graph $\mathcal{D} = (\mathcal{V}, \mathcal{E})$, we construct an instance of MCT-DECISION with a complete digraph $\mathcal{G}_c = (\mathcal{V}, \mathcal{V} \times \mathcal{V})$, a real number $\tau_0 = \frac{N+2}{N}$ where $N$ is the size of $\mathcal{V}$, and delay function $d_c$, where for a given arbitrary choice of vertex $v_0$, $d_c$ is defined as:

$$d_c(i,j) = \begin{cases} 1 & \text{if } ((i,j) \in \mathcal{E}) \wedge (j \neq v_0) \wedge (i \neq v_0), \\ 2 & \text{if } (((i,j) \in \mathcal{E}) \wedge ((j = v_0) \vee (i = v_0))) \vee (((i,j) \notin \mathcal{E}) \wedge (j \neq v_0) \wedge (i \neq v_0)), \\ 3 & \text{if } ((i,j) \notin \mathcal{E}) \wedge ((j = v_0) \vee (i = v_0)). \end{cases}$$

The constructed digraph $\mathcal{G}_c$ is complete and the delays are symmetric and verify triangular inequality. In fact for three distinct nodes $i, j$, and $k$ in $\mathcal{V}$, we prove that $d_c(i,j) \leq d_c(i,k) + d_c(k,j)$ by distinguishing three possible cases:

1. If $i \neq v_0$ and $j \neq v_0$, then $d_c(i,j) \leq 2$, but every delay is at least equal to one and then $2 \leq d_c(i,k) + d_c(k,j)$; it follows that $d_c(i,j) \leq d_c(i,k) + d_c(k,j)$.

2. If $i = v_0$, then $d_c(v_0,k) \geq 2$, thus $d_c(v_0,k) + d_c(k,j) \geq 3$. It follows that $d_c(v_0,j) \leq 3 \leq d_c(v_0,k) + d_c(k,j)$.

3. The case when $j = v_0$ is analogous to the case when $i = v_0$.

If $\mathcal{D}$ has a Hamiltonian cycle, then the (directed) graph induced by this cycle is a strong spanning subdigraph of $\mathcal{G}_c$ and its cycle time is $\tau_{\text{HC}} = \frac{1 \times (N-2) + 2 + 2}{N} = \frac{N+2}{N} \leq \tau_0$.

If $\mathcal{G}_c$ has a strong spanning sub-digraph, say $\mathcal{G}^*$, having a cycle time $\tau^* \leq \frac{N+2}{N}$, let $C$ be an elementary circuit of $\mathcal{G}^*$ containing $v_0$ (such a circuit always exists because the graph is strongly connected). By definition of cycle time, $\frac{d_c(C)}{|C|} \leq \tau^* = 1 + \frac{2}{N}$. We are going to prove that $C$ is a Hamiltonian cycle of $\mathcal{D}$.

We prove first by contradiction that $C$ contains only the arcs from $\mathcal{E}$. Suppose by contradiction that there exists an arc $(i,j) \notin \mathcal{E}$ in $C$, two cases are possible:

1. If $j \neq v_0$, and $i \neq v_0$ then $d_c(i,j) = 2$ and since $v_0 \in C$, there exist two nodes $v_0^- \in \sigma(v_0)$ and $v_0^+ \in \pi(v_0)$ in $C$. It follows that $d_c(C) \geq d_c(i,j) + d(v_0^+, v_0) + d_c(v_0, v_0^-) + 1 \times (|C| - 3) \geq 2 + 2 + 2 + |C| - 3 = |C| + 3$. Since $C$ is an elementary circuit, it follows that $|C| \leq N$, thus $\frac{d_c(C)}{|C|} \geq 1 + \frac{3}{N}$, and this contradicts $\frac{d_c(C)}{|C|} \leq 1 + \frac{2}{N}$.

2. If $i = v_0$, let $v_0^+$ be the predecessor of $v_0$ in $C$, it follows that $d_c(C) \geq d_c(v_0^+, v_0) + d(v_0, j) + 1 \times (|C| - 2) \geq 3 + 2 + |C| - 2 = 3 + |C|$, thus $\frac{d_c(C)}{|C|} \geq 1 + \frac{3}{|C|}$, and using the same argument as for the first case we get a contradiction.

3. The case when $j = v_0$ is analogous to the case when $i = v_0$.

It follows that any arc of $C$ is in $\mathcal{E}$.

We prove next that $C$ is a Hamiltonian Cycle, i.e., $|C| = N$. Since $v_0 \in C$, there exist two nodes $v_0^+ \in \sigma(v_0)$ and $v_0^- \in \pi(v_0)$ in $C$, it follows that $d_c(C) = d_c(v_0^-, v_0) + d_c(v_0, v_0^+) + 1 \times (|C| - 2) = 2 + 2 + |C| - 2 = 2 + |C|$.

Since $\frac{d_c(C)}{|C|} \leq \tau^* = 1 + \frac{2}{N}$, it follows that $1 + \frac{2}{|C|} \leq 1 + \frac{2}{N}$, thus $|C| \geq N$. As $C$ is an elementary circuit it follows that $|C| = N$, i.e., $C$ is a Hamiltonian cycle. Since $C$ is a circuit containing only arcs from $\mathcal{D}$, it follows that $\mathcal{D}$ has a Hamiltonian cycle.

So we have proved that $\mathcal{D}$ has a Hamiltonian cycle if and only if $\mathcal{G}_c$ has strong spanning subdigraph of cycle time at most $\tau_0 = \frac{N+2}{N}$. It follows that MCT-DECISION is NP-complete, thus MCT is NP-hard even when $\mathcal{G}_c$ is a complete Euclidean edge-capacitated graph. $\qquad\square$

### E.3  Proof of Proposition 3.3

Under the assumption that the connectivity topology is Euclidean (delays are symmetric and verify triangular inequality), we first show that the solution of Travelling Salesman Problem (TSP) [33] is guaranteed to be within a $2N$-multiplicative factor of the solution of MCT (Lemma E.3). As a result, the Christofides algorithm [73] which is a 1.5-approximation algorithm for TSP, is a $3N$-approximation algorithm for MCT (Prop. 3.3).

**Lemma E.3.** *Consider an Euclidean digraph $\mathcal{G}_c$ with $N$ nodes and let $\mathcal{H}^*$ denote its optimal tour. Then $\frac{d_c(\mathcal{H}^*)}{|\mathcal{H}^*|} \leq 2N \times \tau_*$, where $\tau_*$ is the optimal cycle time that can be achieved by a strong spanning subdigraph of $\mathcal{G}_c$.*

*Proof.* Let $\mathcal{G}^*$ be a spanning digraph of $\mathcal{G}_c$ with optimal cycle time $\tau^*$.

Let $\{\mathcal{C}_i\}_{i=1,\dots,c}$ be a minimal set of elementary circuits of $\mathcal{G}_*$, so that $\cup_{i=1}^c \mathcal{C}_i = \mathcal{G}_*$ and $\cup_{i \neq j} \mathcal{C}_i \neq \mathcal{G}_*$ for each $j$ (as illustrated in Fig. 6a). Consider an auxiliary graph $\mathcal{G}'$ whose $c$ nodes represent the $c$ circuits and whose links correspond to two circuits sharing a node. Let $\mathcal{T}$ be a spanning tree of $\mathcal{G}'$. Starting from the root of $\mathcal{T}$, we can define an order of the nodes in each circuit and an order of the children of each circuit as follows. Given the orientation of the circuit corresponding to the root, consider the first node they share with each child. We order the children according to such order (solving arbitrarily possible ties). For each child we reorder its nodes starting from the node they share with the father and following the orientation of the circuit. We consider then the ordered traversal of the circuits $\Gamma = (\mathcal{C}_{i_1}, \mathcal{C}_{i_2}, \dots, \mathcal{C}_{i_{2c+1}} = \mathcal{C}_{i_1})$ obtained using DFS on $\mathcal{T}$ and visiting the children according to the order introduced above (as illustrated in Fig. 6b).

From $\Gamma$ we can build two closed walks $\mathcal{W}_1$ and $\mathcal{W}_2$, both spanning all nodes of $\mathcal{G}^*$. The walk $\mathcal{W}_1$ is built by considering all circuits in the order they appear in $\Gamma$, and then concatenating their nodes as follows. The first time we visit one circuit we take all nodes in the circuit in their order (but the last one in each circuit that coincides with the first one). When we come back to the circuit, we only pick the nodes needed to move to the following circuit in $\Gamma$. The walk $\mathcal{W}_2$ is built by considering the $c$ circuits in the order they first appear in $\Gamma$, and then again concatenating their nodes (but the last one in each circuit that coincides with the first one). Both sequences of nodes define walks as $\mathcal{G}_c$ is Euclidean and then complete. The length of $\mathcal{W}_2$ is $|\mathcal{W}_2| = \sum_{i=1}^c |\mathcal{C}_i| \leq N^2$, as we can have at most $N-1$ elementary circuits and each of them has length at most $N$. See Figs. 6c and 6d for the examples of $\mathcal{W}_1$ and $\mathcal{W}_2$.

(a) Circuits decomposition

(b) Nodes ordering

(c) Walk $\mathcal{W}_1$

(d) Walk $\mathcal{W}_2$

Figure 6: Illustration of building walks used in the proof of Lemma E.3.

We observe that $d_c(\mathcal{W}_1) \leq 2 \sum_{i=1}^{c} d_c(\mathcal{C}_i)$ as the walk $\mathcal{W}_1$ passes through each link in each circuit $\mathcal{C}_i$ at most twice: it walks through the first $|\mathcal{C}_i| - 1$ edges of $\mathcal{C}_i$ the first time it visits $\mathcal{C}_i$, and uses once more the edges in $\mathcal{C}_i$ to visit the other circuits and go back to the root. As $\mathcal{W}_2$ is a sublist of the nodes in $\mathcal{W}_1$ and delays satisfy the triangle inequality, it holds $d_c(\mathcal{W}_2) \leq d_c(\mathcal{W}_1)$.

Finally, from the walk $\mathcal{W}_2$ we can extract a Hamiltonian cycle $\mathcal{H}$ that has an even smaller delay. Let $\mathcal{H}^*$ be an optimal tour. It follows

$$\tau(\mathcal{H}^*) = \frac{d_c(\mathcal{H}^*)}{|\mathcal{H}^*|} \leq \frac{d_c(\mathcal{H})}{|\mathcal{H}^*|} \leq \frac{d_c(\mathcal{W}_2)}{|\mathcal{H}^*|} \tag{11}$$

$$= \frac{|\mathcal{W}_2|}{|\mathcal{H}^*|} \frac{d_c(\mathcal{W}_2)}{|\mathcal{W}_2|} \tag{12}$$

$$\leq \frac{N^2}{N} \frac{d_c(\mathcal{W}_1)}{\sum_{i=1}^{c} |\mathcal{C}_i|} \tag{13}$$

$$\leq 2N \frac{\sum_{i=1}^{c} d_c(\mathcal{C}_i)}{\sum_{i=1}^{c} |\mathcal{C}_i|} \tag{14}$$

$$\leq 2N \max_{i=1,\dots,c} \frac{d_c(\mathcal{C}_i)}{|\mathcal{C}_i|} = 2N\tau^*. \tag{15}$$

$\square$

**Proposition 3.3.** *Christofides' algorithm [73] is a $3N$-approximation algorithm for* MCT *when* $\mathcal{G}_c$ *is edge-capacitated and Euclidean.*

*Proof.* Christofides algorithm provides a $\frac{3}{2}$-approximation for the traveling salesman problem TSP defined in [4].[11] Given an instance of MCT let $\hat{C}$ denote the output of Christofides algorithm and $C^*$ denote the optimal tour of $\mathcal{G}_c$. It follows that $d_c(\hat{C}) \leq \frac{3}{2} d_c(C^*)$. Since both $\hat{C}$ and $C^*$ are

Hamiltonian cycles, $|\hat{C}| = |C^*|$. Using Lemma E.3. it follows that $\frac{d_c(\hat{C})}{|\hat{C}|} \le 2N \times \frac{3}{2} \times \tau_* = 3N \times \tau_*$.

Thus the graph obtained using only the edges of $\hat{C}$ is a $3N$-approximation of the MCT problem when $\mathcal{G}_c$ is edge-capacitated and Euclidean. $\qquad\square$

**Observation E.4.** *Christofides' algorithm [73] is a $\Omega(N)$-approximation algorithm for* MCT *when $\mathcal{G}_c$ is edge-capacitated and Euclidean.*

*Proof.* Christofides' algorithm returns a ring as solution. We provide an example of an Euclidean underlay where any ring has cycle time at least $N/4$ times larger than the optimal overlay. We consider a complete connectivity graph $\mathcal{G}_c = (\mathcal{V}, \mathcal{V} \times \mathcal{V})$ to which we associate a delay function $d_c$ verifying

$$\forall (i,j) \in \mathcal{V} \times \mathcal{V}; \; d_c(i,j) = \begin{cases} 0 & \text{if } i,j \in \{1, \ldots, N\}, \\ 1 & \text{if } i \in \{N+1, \ldots, 2N\} \text{ or } j \in \{N+1, \ldots, 2N\}. \end{cases} \quad (16)$$

$\mathcal{G}_c$ is clearly an Euclidean graph.

A Hamiltonian cycle $\mathcal{H}$ of $\mathcal{G}_c$ needs to use exactly $2N$ different edges and in particular $N$ different edges with delay 1 to connect nodes $i \in \{N+1, \ldots, 2N\}$. Therefore, the total delay of the cycle is at least $N \times 0 + N \times 1 = N$, and its cycle time $\tau(\mathcal{H}) \ge \frac{N}{2N} = \frac{1}{2}$.

Consider a directed overlay $\mathcal{G}_o = (\mathcal{V}, \mathcal{E}_o)$, with

$$\mathcal{E}_o = \{(i, i+1); \; i \in \{1, \ldots, N-1\}\} \cup \bigcup_{K \in \{N+1, \ldots, 2N\}} \{(N, K), (K, 1)\}. \quad (17)$$

The set of elementary circuits of $\mathcal{E}_o$ is exactly the set

$$\mathcal{C} = \{C_K = (1, \ldots, N, K, 1) : K \in \{N+1, 2N\}\}.$$

For any circuit $C_K \in \mathcal{C}$,

$$\tau(C_K) = \frac{0 \times (N-1) + 2 \times 1}{N+1} = \frac{2}{N+1}.$$

It follows that the minimal cycle time $\tau_{\text{OPT}} = \frac{2}{N+1}$, and $\tau(\mathcal{H}) \ge \frac{N+1}{4}\tau_{\text{OPT}}$ for any Hamiltonian cycle $\mathcal{H}$ of $\mathcal{G}_c$. $\qquad\square$

### E.4 Proof of Proposition 3.4

We prove that in a node-capacitated network, MCT is NP-hard even when $\mathcal{G}_o$ is required to be undirected. We start introducing the associated decision problem:

MCT-U-Decision
**Input:** A strongly connected directed graph $\mathcal{G}_c = (\mathcal{V}, \mathcal{E}_c)$, model size $M$, $\{C_{\text{UP}}(i), C_{\text{DN}}(j), l(i,j), A(i',j'), T_c(i), \forall(i,j) \in \mathcal{E}_c\}$, and a constant $\tau_0 > 0$.
**Output:** Is there a strong spanning undirected subgraph $\mathcal{G}_o$ of $\mathcal{G}_c$, such that $\tau(\mathcal{G}_o) \le \tau_0$?

MCT-U-Decision is closely related to the *degree-constrained spanning tree* (DCST) defined below:

Degree-constrained spanning tree (DCST)
**Input:** An $N$-node connected undirected graph $\mathcal{G} = (\mathcal{V}, \mathcal{E})$; positive integer $k \le N$.
**Output:** Does $\mathcal{G}$ have a spanning tree in which no node has degree greater than $k$?

DCST is a simpler version of $\delta$-MBST, where we look for a spanning tree with degree at most $k$ and minimum bottleneck.

DCST is NP-complete [27]. For example for $k = 2$ it can be shown by a reduction from HC.

**Proposition 3.4.** *In node-capacitated networks* MCT *is NP-hard even when the overlay is required to be undirected.*

*Proof.* Our proof is based on a reduction of DCST to MCT-U-Decision.

Given an instance of DCST with an $N$-node connected undirected graph $\mathcal{G} = (\mathcal{V}, \mathcal{E})$ and a positive integer $k \leq N$, we define an instance of MCT-U-Decision on a connected graph $\mathcal{G}_c = (\mathcal{V}_c, \mathcal{E}_c)$ built from $\mathcal{G}$ according to the following mapping $\Pi$: For each node $v$ in $\mathcal{V}$, there are two nodes $v^{(1)}$ and $v^{(2)}$ in $\mathcal{V}_c$ and $(v^{(1)}, v^{(2)}) \in \mathcal{E}_c$, and for an arc $(v_i, v_j) \in \mathcal{E}$, there is an arc $(v_i^{(1)}, v_j^{(1)})$ in $\mathcal{E}_c$. We set $\frac{M}{C_{\text{UP}}(v^{(1)})} = 1$, $\frac{M}{C_{\text{UP}}(v^{(2)})} = k+1$ for all $v \in \mathcal{V}$, $T_c(i) = 0$, $C_{DN}(i) = \infty$ for all $i \in \mathcal{V}_c$, and $l(i,j) = 0$, $A(i', j') = \infty$ for all $(i,j) \in \mathcal{E}_c$. Finally, we consider $\tau_0 = k+1$.

Suppose that $\mathcal{G}$ has a spanning tree $\mathcal{T} = (\mathcal{V}, \mathcal{E}_{\mathcal{T}})$ in which no node has degree greater than $k$, and denote $\mathcal{T}_c = \Pi(\mathcal{T})$ (i.e., we apply the same mapping described above). $\mathcal{T}_c$ is a spanning tree of $\mathcal{G}_c$ (it is acyclic and spans all nodes of $\mathcal{G}_c$). All elementary circuits of $\mathcal{T}_c$ are either of the form $(v_i^{(1)}, v_i^{(2)}, v_i^{(1)})$ for some $v_i \in \mathcal{V}$, or of the form $(v_i^{(1)}, v_j^{(1)}, v_i^{(1)})$ for some $(v_i, v_j) \in \mathcal{E}_{\mathcal{T}}$. Moreover, $\tau((v_i^{(1)}, v_i^{(2)}, v_i^{(1)})) = \frac{k+1+\text{degree}_{\mathcal{T}}(v_i)+1}{2} \leq k+1$ and $\tau((v_i^{(1)}, v_j^{(1)}, v_i^{(1)})) = \frac{\text{degree}_{\mathcal{T}}(v_i)+1+\text{degree}_{\mathcal{T}}(v_j)+1}{2} \leq k+1$. It follows that $\tau(\mathcal{T}_c) \leq k+1 = \tau_0$.

Inversely, suppose that $\mathcal{G}_c$ has an MST $\mathcal{T}_c$ having a cycle time at most $\tau_0$, and let $\mathcal{T} = \Pi^{-1}(\mathcal{T}_c)$, where $\Pi^{-1}(\mathcal{T})$ is obtained by deleting all the vertices of the form $v_i^{(2)}$ for $v_i \in \mathcal{V}$. $\mathcal{T}$ is a spanning tree of $\mathcal{G}$ (it contains all nodes of $\mathcal{G}$ and is acyclic). We prove by contradiction that $\text{degree}(\mathcal{T}) \leq k$. Suppose that there exists a node $v \in \mathcal{V}$ such that $|\mathcal{N}_v^-(\mathcal{T})| > k$, it follows that circuit $\{v_i^{(1)}, v_i^{(2)}, v_i^{(1)}\}$ is a circuit of $\mathcal{T}_c$, and $\tau((v_i^{(1)}, v_i^{(2)}, v_i^{(1)})) = \frac{k+1+|\mathcal{N}_v^-(\mathcal{T})|+1}{2} > k+1$. It follows that $\tau(\mathcal{T}_c) > k+1$, thus $k+1 < \tau_0 = k+1$ (contradiction).

Then the answer to DCST is positive if and only if the answer to MCT-U-Decision is positive. In addition, we have a polynomial reduction algorithm. It follows that MCT-U-Decision is NP-hard. $\square$

### E.5  Proof of Proposition 3.5

The bottleneck of a tree $\mathcal{T}$ is its maximum edge weight, denoted by $B(\mathcal{T})$. To prove Prop. 3.5, we start by proving that the bottleneck of the MBST of the undirected graph $\mathcal{G}_c^{(u)}$ (considered in lines 1-3 of Algo. 1) is smaller than or equal to the minimal cycle time of the connectivity graph $\mathcal{G}_c$.

We consider a node-capacitated case where $C_{\text{UP}}(i) \leq \min\left(\frac{C_{\text{DN}}(j)}{N}, A(i', j')\right), \forall (i,j) \in \mathcal{E}_c$. Thus, according to (3), the overlay $\mathcal{G}_o$ has weights

$$d_o(i,j) = s \times T_c(i) + l(i,j) + \frac{M|\mathcal{N}_i^-|}{C_{\text{UP}}(i)}, \quad \forall (i,j) \in \mathcal{E}_c. \tag{18}$$

Note that the weights defined for the undirected graph $\mathcal{G}_c^{(u)} = (\mathcal{V}, \mathcal{E}_c^{(u)})$ are

$$d_c^{(u)}(i,j) = \frac{s \times (T_c(i) + T_c(j)) + l(i,j) + l(j,i) + \frac{M}{C_{\text{UP}}(i)} + \frac{M}{C_{\text{UP}}(j)}}{2}, \quad \forall (i,j) \in \mathcal{E}_c^{(u)}. \tag{19}$$

**Lemma E.5.** *Consider the case where $\mathcal{G}_c$ is node-capacitated with $C_{UP}(i) \leq \min\left(\frac{C_{DN}(j)}{N}, A(i', j')\right)$, $\forall (i,j) \in \mathcal{E}_c$, and the overlay is required to be undirected. Let $\tau^*(\mathcal{G}_c)$ be the cycle time of MCT on $\mathcal{G}_c$ and $\mathcal{T}_{MBST}(\mathcal{G}_c^{(u)})$ be the MBST of $\mathcal{G}_c^{(u)}$. The bottleneck of $\mathcal{T}_{MBST}(\mathcal{G}_c^{(u)})$ is smaller than or equal to $\tau^*(\mathcal{G}_c)$, i.e. $B(\mathcal{T}_{MBST}(\mathcal{G}_c^{(u)})) \leq \tau^*(\mathcal{G}_c)$.*

*Proof.* Denote $\mathcal{T}^*(\mathcal{G}_c)$ the undirected overlay of $\mathcal{G}_c$ with minimal cycle time. We consider the edge

$$(w, v) = \underset{(i,j) \in \mathcal{E}(\mathcal{T}^*(\mathcal{G}_c))}{\arg\max} d_c^{(u)}(i,j).$$

By definition, $B(\mathcal{T}_{MBST}(\mathcal{G}_c^{(u)})) = \min_{\mathcal{T} \in ST(\mathcal{G}_c^{(u)})} \max_{(i,j) \in \mathcal{E}(\mathcal{T})} d_c^{(u)}(i,j)$, where $ST(\mathcal{G}_c^{(u)})$ is the set of spanning trees of $\mathcal{G}_c^{(u)}$. Since $\mathcal{T}^*(\mathcal{G}_c) \in ST(\mathcal{G}_c^{(u)})$, we have:

$B(\mathcal{T}_{MBST}(\mathcal{G}_c^{(u)})) \leq d_c^{(u)}(w, v)$

$$\overset{(19)}{=} \frac{s \times (T_c(w) + T_c(v)) + l(w, v) + l(v, w) + M/C_{\text{UP}}(w) + M/C_{\text{UP}}(v)}{2}$$

$$\leq \frac{s \times (T_c(w) + T_c(v)) + l(w, v) + l(v, w) + |\mathcal{N}_w^-|M/C_{\text{UP}}(w) + |\mathcal{N}_v^-|M/C_{\text{UP}}(v)}{2}$$

$$\overset{(18)}{=} \frac{d_o(w, v) + d_o(v, w)}{2}$$

$$\leq \tau^*(\mathcal{G}_c),$$

where the second inequality follows from $|\mathcal{N}_w^-|, |\mathcal{N}_v^-| \geq 1$, and the last inequality comes from the definition of cycle time. $\square$

Lemma E.5 establishes a connection between the bottleneck of the MBST of $\mathcal{G}_c^{(u)}$ and the cycle time of MCT on $\mathcal{G}_c$ when the overlay is required to be undirected. To get an approximated 2-MBST on $\mathcal{G}_c^{(u)}$, we apply the best known 3-approximation algorithm from [3, Sect. 3.2.1] (see lines 6-8 in Algo. 1) which requires $\mathcal{G}_c^{(u)}$ to be Euclidean. So in the following, we show that indeed $\mathcal{G}_c^{(u)}$ is Euclidean.

**Lemma E.6.** *If $\mathcal{G}_c$ is Euclidean, then $\mathcal{G}_c^{(u)}$ is Euclidean.*

*Proof.* Remind that the connectivity graph $\mathcal{G}_c$ is Euclidean on a node-capacitated network, if its delays $d_c(i, j) = s \times T_c(i) + l(i, j)$ are symmetric ($d_c(i, j) = d_c(j, i), \forall i, j \in \mathcal{V}$) and satisfy the triangle inequality. From (19) it is easy to check that $d_c^{(u)}(i, j) = d_c^{(u)}(j, i)$. Consider three nodes $i, j, k \in \mathcal{V}$, we have:

$$d_c^{(u)}(i, j) = \frac{d_c(i, j) + d_c(j, i) + M/C_{\text{UP}}(i) + M/C_{\text{UP}}(j)}{2}$$

$$\leq \frac{d_c(i, k) + d_c(k, j) + d_c(j, k) + d_c(k, i) + M/C_{\text{UP}}(i) + M/C_{\text{UP}}(j)}{2}$$

$$\leq \frac{d_c(i, k) + d_c(k, j) + d_c(j, k) + d_c(k, i) + M/C_{\text{UP}}(i) + M/C_{\text{UP}}(j) + 2M/C_{\text{UP}}(k)}{2}$$

$$= d_c^{(u)}(i, k) + d_c^{(u)}(k, j),$$

where the first inequality follows from the triangle inequality for $d_c(i, j)$ and the second inequality from $C_{\text{UP}}(k) \geq 0$. $\square$

**Proposition 3.5.** *Algorithm 1 is a 6-approximation algorithm for MCT when $\mathcal{G}_c$ is node-capacitated and Euclidean with $C_{\text{UP}}(i) \leq \min\left(\frac{C_{\text{DN}}(j)}{N}, A(i', j')\right), \forall (i, j) \in \mathcal{E}_c$, and $\mathcal{G}_o$ is required to be undirected.*

*Proof.* Algorithm 1 considers, as a candidate solution, an opportune Hamiltonian path $\mathcal{H}$ (line 8) for which reference [3, Thm. 8] proves that

$$B(\mathcal{H}) \leq 3 \times B(\mathcal{T}_{MBST}(\mathcal{G}_c^{(u)})) \tag{20}$$

as $\mathcal{G}_c^{(u)}$ is Euclidean (Lemma E.6). Moreover,

$$\tau(\mathcal{H}) = \max_{(i,j) \in \mathcal{E}(\mathcal{H})} \frac{d_o(i, j) + d_o(j, i)}{2}$$

$$= \max_{(i,j) \in \mathcal{E}(\mathcal{H})} \frac{s \times T_c(i) + s \times T_c(j) + l(i, j) + l(j, i) + \frac{M|\mathcal{N}_i^-|}{C_{\text{UP}}(i)} + \frac{M|\mathcal{N}_j^-|}{C_{\text{UP}}(j)}}{2}$$

$$\leq \max_{(i,j) \in \mathcal{E}(\mathcal{H})} \frac{s \times T_c(i) + s \times T_c(j) + l(i, j) + l(j, i) + 2\frac{M}{C_{\text{UP}}(i)} + 2\frac{M}{C_{\text{UP}}(j)}}{2}$$

$$\leq \max_{(i,j) \in \mathcal{E}(\mathcal{H})} s \times T_c(i) + s \times T_c(j) + l(i, j) + l(j, i) + \frac{M}{C_{\text{UP}}(i)} + \frac{M}{C_{\text{UP}}(j)}$$

$$= 2 \max_{(i,j) \in \mathcal{E}(\mathcal{H})} d_c^{(u)}(i, j)$$

$$= 2B(\mathcal{H}), \tag{21}$$

where the first inequality follows from nodes in a path having degree at most 2. Combining (20), (21), and Lemma E.5, it follows that $\tau(\mathcal{H}) \leq 6 \times \tau^*(\mathcal{G}_c)$. □

### E.6 Proof of Proposition 3.6

**Proposition 3.6.** *Christofides' algorithm is a $3N$-approximation algorithm for* MCT *when* $\mathcal{G}_c$ *is node-capacitated and Euclidean.*

*Proof.* Let $\mathcal{G}'_c$ be a weighted graph with the same topology as $\mathcal{G}_c$ with weights $d'(i,j) = s \times T_c(i) + l(i,j) + \frac{M}{\min(C_{\text{UP}}(i), C_{\text{DN}}(j), A(i',j'))}$. Denote $\hat{C}$ the output of Christofides' algorithm when used on $\mathcal{G}'_c$, and denote $C^*$ the optimal tour of $\mathcal{G}'_c$. Since Christofides' algorithm provides a $\frac{3}{2}$-approximation to TSP, it follows that $d'(\hat{C}) \leq \frac{3}{2}d'(C^*)$. As $\hat{C}$ and $C^*$ are directed rings, it holds $d'(\hat{C}) = d_o(\hat{C})$ and $d'(C^*) = d_o(C^*)$. Using Lemma E.3 it follows that

$$\tau(\hat{C}) = \frac{d_o(\hat{C})}{|\hat{C}|} = \frac{d'(\hat{C})}{|\hat{C}|} \leq \frac{3}{2}\frac{d'(C^*)}{|C^*|} = \frac{3}{2}\frac{d_o(C^*)}{|C^*|} = \frac{3}{2}\tau(C^*) \leq 3N\tau^*.$$

Thus the graph obtained using only the edges of $\hat{C}$ is a $3N$-approximation algorithm for MCT when $\mathcal{G}_c$ is node-capacitated and Euclidean. □

# F   Time Simulator

The time simulator reconstructs the wall-clock time. It requires the complete knowledge about the underlay topology, i.e., the capacities of all physical links and the upload and download capacities for each silo. For a given overlay topology $\mathcal{G}_o = (\mathcal{V}, \mathcal{E}_o)$, the purpose of the proposed time simulator (Alg. 3) is to compute $t(k) = (t_i(k))_{1 \le i \le N}$, i.e., the time at which each silo starts computing for the $k$-th time. The simulator needs to compute the delay required to send a message with a known size on each physical link of the underlay. This delay is the sum of two terms [59]:

- Latency: it is the time required by the first transmitted bit to travel from the source to the destination. The latency of a link $(i, j)$ essentially depends on the length of the link and the speed of the light in the link's transmission medium. We have estimated the latency using the formula proposed in [32]: $0.0085 \times \text{distance}(i, j) + 4$, where the distance is expressed in kilometers and the latency in milliseconds. The latency of a path is the sum of the link latencies.

- Transmission Delay: it is the time between the reception of the first bit of the message and the reception of the last bit. It depends on the minimum available bandwidth along the path. We compute it as $M / \min \left( \frac{C_{\text{UP}}(i)}{|\mathcal{N}_i^-|}, \frac{C_{\text{DN}}(j)}{|\mathcal{N}_j^+|}, A(i', j') \right)$.

Finally, the simulator also accounts for the total time spent in computation by each node, that is the product of the number of local steps $s$ and the time needed to perform one local step (in milliseconds), i.e., $s \times T_c(i)$.

---

**Algorithm 3:** Time Simulator

**Input :** $(l_{i,j})_{(i,j) \in \mathcal{G}_o}$, $(T_i^c)_{i \in \mathcal{V}}$, $(C_{\text{DN}}(i))_{i \in \mathcal{V}}$ and $(C_{\text{UP}}(i))_{i \in \mathcal{V}}$
**Result:** $t \in \mathbb{R}^{N \times K}$

1 **for** $i \in \mathcal{V}$ **do**
2 $\quad$ $t_i(0) = 0$;
3 **end**
4 **for** $k \in \{1, \dots, K\}$ **do**
5 $\quad$ $t_i(k) = \max_{j \in \mathcal{N}_i^+} \left( t_j(k-1) + l(i,j) + \dfrac{M}{\min \left( \frac{C_{\text{UP}}(i)}{|\mathcal{N}_i^-|}, \frac{C_{\text{DN}}(j)}{|\mathcal{N}_j^+|}, A(i',j') \right)} \right)$;
6 $\quad$ $t_i(k) = t_i(k) + s \times T_c(i)$;
7 **end**

---

(a) Available bandwidth between some pairs of silos in Géant as computed through our model.

(b) Available bandwidth measurements between Gaia sites [38, Fig. 2].

Figure 7: Our simulator with 1 Gbps capacity links generates a distribution of available bandwidths with the same variability observed in real networks.

(a) Underlay

(b) Star

(c) MST

(d) Ring

Figure 8: Géant Network: the underlay (a) and selected overlays computed when core links have 1 Gbps capacity and access links have 10 Gbps capacity (b-d).

# G  Experiments Detailed Description

## G.1  Networks and Communication model

We considered three real topologies from *Rocketfuel engine* [94] (Exodus and Ebone) and from *The Internet Topology Zoo* [48] (Géant), and two synthetic topologies (AWS North-America and Gaia) built from AWS data centers [38, 96] (Table 3). For the synthetic topologies, we consider a full-meshed underlay. We assume all underlays support a shortest path routing with the geographical distance (or equivalently the latency) as link cost. These topologies have between 11 and 87 nodes located in the same continent with the exception of Gaia, which spans four continents. The Géant and Ebone network connect European cities and Exodus network connect American cities. We considered that each network node is connected to a geographically close silo by a symmetric access link.

Some underlays and examples of overlays are shown in Figures 8, 9, and 10.

## G.2  Datasets and Models

We provide full details on datasets and models used in our experiments. We use multiple datasets spanning a wide range of machine learning tasks (sentiment analysis, language modeling, image classification, handwritten character recognition), including those used in prior work on federated learning [72], and in LEAF [14] benchmark, and a cross-silo specific dataset based on iNaturalist [99].

**iNaturalist dataset.**    iNaturalist [99] consists of images from over 8,000 different species of plants and animals. We choose the dataset from iNaturalist 2018 competition which contains 450,000

|   (a) Underlay   |   (b) Star   |   (c) MST   |   (d) Ring   |

Figure 9: Gaia Network: the underlay (a) and selected overlays computed when core links have 1 Gbps capacity and access links have 10 Gbps capacity (b-d).

|   (a) Underlay   |   (b) Star   |   (c) MST   |   (d) Ring   |

Figure 10: AWS-North America Network: the underlay (a) and selected overlays computed when core links have 1 Gbps capacity and access links have 10 Gbps capacity (b-d).

images[12] where the geo-locations of these images are provided. Due to a large class imbalance, iNaturalist species classification is a tough learning task, which requires large computation resources. In our experiments, we started by using a subset of the original iNaturalist dataset, selecting images containing the 80 most popular species.[13] We have also conducted additional experiments on the full iNaturalist dataset, whose corresponding results are presented in Appendix H.4. We refer to the complete dataset as Full-iNaturalist.

In order to simulate a realistic cross-silo environment with non-iid local datasets, one can assign the images to the geographically closest silo obtaining local datasets different in size and in the species represented. This distribution would lead some silos to have no point. We decided then to assign half of the images uniformly at random and half to the closest silo. Moreover, since most of the images in iNaturalist are from North America, for European networks such as Ebone and Géant, we mapped the European cities westward by reducing their longitude by 90 degrees. Table 4 shows that our method generates quite unbalanced data distribution (e.g., for Ebone, one silo can have up to 50 times more images than another one).

To classify iNaturalist images we finetuned a pretrained ResNet-18 on ImageNet [22]. In particular we used the torchvision [70] implementation of ResNet-18.

**LEAF datasets.** LEAF [14] is a benchmark framework for learning in federated settings. We used three LEAF datasets in our experiments on AWS North America network where we took $20\%$ of the samples randomly as our dataset.[14] Statistics for the corresponding data distributions are in Table 5.

- **FEMNIST** (*Federated Extended* MNIST): A 62-class image classification dataset built by partitioning the data of Extended MNIST based on the writer of the digits/characters. In our experiments, we associate each silo with a random number of writers following a lognormal distribution with mean equal to 5 and standard deviation equal to 1.5.

  We train a convolutional neural network, similar to LeNet, with two convolutional layers followed by a max-pooling layer and two fully connected layers.

Table 4: Statistics of iNaturalist dataset distribution for different networks.

| Network name | Silos | Samples/silo | | | |
| --- | --- | --- | --- | --- | --- |
| | | Mean | Stdev | Min | Max |
| Gaia | 11 | 1213 | 1143 | 610 | 3981 |
| AWS North America | 22 | 606 | 731 | 113 | 3216 |
| Géant | 40 | 333 | 644 | 152 | 4261 |
| Exodus | 79 | 168 | 96 | 92 | 576 |
| Ebone | 87 | 153 | 394 | 68 | 3389 |

Table 5: Statistics of LEAF dataset distribution for AWS North America network (22 silos).

| Dataset | Samples/silo | | | |
| --- | --- | --- | --- | --- |
| | Mean | Stdev | Min | Max |
| Shakespeare | 36359 | 6837 | 24207 | 50736 |
| FEMNIST | 6847 | 7473 | 196 | 26469 |
| Sentiment140 | 13101 | 14273 | 424 | 50562 |

- **Shakespeare**: A dataset built from *The Complete Works of William Shakespeare*, which is partitioned by the speaking roles [72]. In our experiment, we associate each silo with a random number of speaking roles following a lognormal distribution with mean equal to 5 and standard deviation equal to 1.5.

  We consider character-level based language modeling on this dataset. The model takes as input a sequence of 200 English characters and predicts the next character. The model embeds the 200 characters into a learnable 16 dimensional embedding space, and uses two stacked-GRU layers with 256 hidden units, followed by a densely-connected layer.

- **Sentiment140** [30]: An automatically generated sentiment analysis dataset that annotates tweets based on their emoticons. In our experiment, we associate each silo with a random number of Twitter accounts following a lognormal distribution with mean equal to 5 and standard deviation equal to 1.5.

  We use a two layer bi-directional LSTM binary classifier containing 256 hidden units with pretrained 100 dimensional GloVe embedding [82].

### G.3 Implementation Details

**Machines.** The experiments have been run on a CPU/GPU cluster, with different GPUs available (e.g., Nvidia Tesla V100, GeForce GTX 1080 Ti, and Titan X).

**Libraries.** All code is implemented in PyTorch Version 1.4.0. We offer two possibilities for running the code: *sequential* (using only one GPU) and *parallel* (using multiple GPUs). In the parallel setting MPI backend is used for inter-GPU communications.

**Hyperparameters.** The dataset is randomly split into an $80\%$ training set and a $20\%$ testing set. When training on Gaia, AWS North America, and Géant networks, the initial learning rate is set to $0.001$ with Adam optimizer. When training on Exodus and Ebone networks, the initial learning rate is set to $0.1$ with SGD optimizer. We decay the learning rate based on the inverse square root of the number of communication rounds. The batch size is set to $512$ for Sentiment140 and Shakespeare datasets, to $128$ for Femnist dataset and to $16$ for iNaturalist dataset.

**Consensus Matrix.** For a given overlay $\mathcal{G}_o = (\mathcal{V}, \mathcal{E}_o)$, the consensus matrix $\boldsymbol{A}$ is selected similarly to the local-degree rule in [62]. The weight on an arc is based on the larger in-degree of its two incident nodes:

$$\boldsymbol{A}_{i,j} = \frac{1}{1 + \max\left(|\mathcal{N}_i^-|, |\mathcal{N}_j^-|\right)}, \quad \forall (i,j) \in \mathcal{E}_o. \tag{22}$$

$$\boldsymbol{A}_{i,i} = 1 - \sum_{j \in \mathcal{N}_i^-} \boldsymbol{A}_{i,j}, \quad \forall i \in \mathcal{V}. \tag{23}$$

The matrix $\boldsymbol{A}$ so-built is symmetric doubly stochastic. The weights can be determined in a fully-distributed way: every node just needs to exchange degree information with its neighbours.

**MATCHA.**  We implemented MATCHA as described in [104] but for one difference. In MATCHA, each matching $i$ is selected independently with some probability $p_i$. With probability $q = \prod_i (1 - p_i)$, no matching is selected and then no communication occurs. This is equivalent to perform a random number of local steps $s$ between two communication rounds. In order to compare fairly the different approaches and isolate the effect of $s$, we fixed $s$ also for MATCHA as follows. Silos perform a given number of local steps $s$ and then, when a communication should occur, matchings are independently sampled until at least one of them is selected. In practice, in our experiments, the probability $q$ was close to 0, so that the two approaches are practically undistinguishable. Finally, we observe that MATCHA computes the matchings coloring an initial topology, but it is not explained how this initial topology is selected. MATCHA and MATCHA$^+$ operate exactly in the same way but starting from two different initial topologies: the connectivity graph $\mathcal{G}_c$ and the underlay $\mathcal{G}_u$, respectively. The silos can easily discover the connectivity graph $\mathcal{G}_c$; reconstructing the underlay is much more complicated. Nevertheless, as MATCHA$^+$ was in general outperforming MATCHA, we showed the results for MATCHA$^+$.

Table 6: iNaturalist training over different networks. 1 Gbps core links capacities, 10 Gbps access links capacities. Five local computation steps.

| Network name | Silos | Links | Cycle time (ms) | | | | | Ring's training speed-up | |
|---|---|---|---|---|---|---|---|---|---|
| | | | STAR | MATCHA$^{(+)}$ | MST | $\delta$-MBST | RING | vs STAR | vs MATCHA$^{(+)}$ |
| Gaia [38] | 11 | 55 | 492.4 | 329.3(329.3) | 239.7 | 239.8 | 219.7 | 1.79 | 1.50(1.50) |
| AWS NA [96] | 22 | 231 | 389.8 | 226.0(226.0) | 191.3 | 191.3 | 182.9 | 1.40 | 1.24(1.24) |
| Géant [29] | 40 | 61 | 736.0 | 553.8(207.4) | 202.6 | 202.6 | 210.6 | 3.49 | 2.63(2.96) |
| Exodus(us) [68] | 79 | 147 | 1013.4 | 695.0(243.8) | 246.9 | 246.9 | 205.5 | 3.95 | 2.25(1.18) |
| Ebone(eu) [68] | 87 | 161 | 1003.2 | 681.6(224.9) | 223.2 | 223.2 | 196.9 | 3.04 | 2.29(1.21) |

Table 7: iNaturalist training over different networks. 1 Gbps core links capacities, 10 Gbps access links capacities. Ten local computation steps.

| Network name | Silos | Links | Cycle time (ms) | | | | | Ring's training speed-up | |
|---|---|---|---|---|---|---|---|---|---|
| | | | STAR | MATCHA$^{(+)}$ | MST | $\delta$-MBST | RING | vs STAR | vs MATCHA$^{(+)}$ |
| Gaia [38] | 11 | 55 | 619.4 | 456.4(456.4) | 366.7 | 366.7 | 346.7 | 1.79 | 1.32(1.32) |
| AWS NA [96] | 22 | 231 | 516.8 | 353.2(353.2) | 318.3 | 318.3 | 309.9 | 0.69 | 0.47(0.47) |
| Géant [29] | 40 | 61 | 609.0 | 680.8(334.7) | 329.6 | 329.6 | 337.6 | 0.90 | 1.00(1.98) |
| Exodus(us) [68] | 79 | 147 | 1140.4 | 822.0(370.9) | 373.9 | 373.9 | 332.5 | 1.52 | 1.10(1.23) |
| Ebone(eu) [68] | 87 | 161 | 1130.2 | 808.6(352.1) | 350.4 | 350.4 | 323.9 | 1.74 | 1.25(1.09) |

# H    Complete Set of Experiments

## H.1    Effect of the number of local steps

Tables 6 and 7 show the effect of 6 different overlays when training ResNet-18 over iNaturalist in networks with 1 Gbps core links and 10 Gbps access links and local steps equal to 5 and 10, respectively. For 5 local steps, the training time is evaluated as the time to reach a training accuracy equal to 65%, 55%, 60%, 45%, and 45% for Gaia, AWS North America, Géant, Exodus, and Ebone, respectively. For 10 local steps, the training time is evaluated as the time to reach a training accuracy equal to 65%, 50%, 50%, 45%, and 40%, respectively.

## H.2    Full results for training every dataset on AWS North America

In Figure 2, we have shown the training loss w.r.t. communication rounds and wall-clock time when training four different datasets on AWS North America. Here we provide the complete results (Figures 11–14) which include training loss, training accuracy, test loss, and test accuracy w.r.t communication rounds and wall-clock time.

## H.3    Exploring other scenarios

In our experiments, we considered 5 underlays, for which we compared 6 different overlays (e.g., Table 3). Moreover, we tested 4 different datasets (e.g., Fig. 2) and 3 different values for the number of local steps $s = 1, 5, 10$ (e.g., Tables 6 and 7). We were not able to run experiments for all 360 possible combinations. In Figures 15–24, we show some representative additional results. For each experimental result, four metrics are shown including the train loss, train accuracy, test loss, and test accuracy w.r.t. communication rounds and wall-clock time. The common observation is that the RING converges faster than MATCHA$^+$ and STAR in terms of wall-clock time. In some cases, the test loss and accuracy of the model learned by the RING start becoming worse after some time, with overfitting being a possible explanation in some cases (see Figs. 15, 17, 20, and 22).

(a) Train Loss  (b) Train Accuracy  (c) Test Loss  (d) Test Accuracy

Figure 11: Effect of overlays on the convergence w.r.t. communication rounds (top row) and wall-clock time (bottom row) when training Shakespeare on AWS North America underlay. 1 Gbps core links capacities, 100 Mbps access links capacities, $s = 1$.

(a) Train Loss  (b) Train Accuracy  (c) Test Loss  (d) Test Accuracy

Figure 12: Effect of overlays on the convergence w.r.t. communication rounds (top row) and wall-clock time (bottom row) when training FEMNIST on AWS North America underlay. 1 Gbps core links capacities, 100 Mbps access links capacities, $s = 1$.

(a) Train Loss  (b) Train Accuracy  (c) Test Loss  (d) Test Accuracy

Figure 13: Effect of overlays on the convergence w.r.t. communication rounds (top row) and wall-clock time (bottom row) when training Sentiment140 on AWS North America underlay. 1 Gbps core links capacities, 100 Mbps access links capacities, $s = 1$.

(a) Train Loss      (b) Train Accuracy      (c) Test Loss      (d) Test Accuracy

Figure 14: Effect of overlays on the convergence w.r.t. communication rounds (top row) and wall-clock time (bottom row) when training iNaturalist on AWS North America underlay. 1 Gbps core links capacities, 100 Mbps access links capacities, $s = 1$.

(a) Train Loss      (b) Train Accuracy      (c) Test Loss      (d) Test Accuracy

Figure 15: Effect of overlays on the convergence w.r.t. communication rounds (top row) and wall-clock time (bottom row) when training ResNet-18 image classification model using iNaturalist on Gaia underlay. 1 Gbps core links capacities, 100 Mbps access links capacities, $s = 1$.

(a) Train Loss      (b) Train Accuracy      (c) Test Loss      (d) Test Accuracy

Figure 16: Effect of overlays on the convergence w.r.t. communication rounds (top row) and wall-clock time (bottom row) when training ResNet-18 image classification model using iNaturalist on AWS North America underlay. 1 Gbps core links capacities, 100 Mbps access links capacities, $s = 1$.

(a) Train Loss  (b) Train Accuracy  (c) Test Loss  (d) Test Accuracy

Figure 17: Effect of overlays on the convergence w.r.t. communication rounds (top row) and wall-clock time (bottom row) when training ResNet-18 image classification model using iNaturalist on Géant underlay. 1 Gbps core links capacities, 100 Mbps access links capacities, $s = 1$.

(a) Train Loss  (b) Train Accuracy  (c) Test Loss  (d) Test Accuracy

Figure 18: Effect of overlays on the convergence w.r.t. communication rounds (top row) and wall-clock time (bottom row) when training ResNet-18 image classification model using iNaturalist on Exodus underlay. 1 Gbps core links capacities, 100 Mbps access links capacities, $s = 1$.

(a) Train Loss  (b) Train Accuracy  (c) Test Loss  (d) Test Accuracy

Figure 19: Effect of overlays on the convergence w.r.t. communication rounds (top row) and wall-clock time (bottom row) when training ResNet-18 image classification model using iNaturalist on Ebone underlay. 1 Gbps core links capacities, 100 Mbps access links capacities, $s = 1$.

(a) Train Loss      (b) Train Accuracy      (c) Test Loss      (d) Test Accuracy

Figure 20: Effect of overlays on the convergence w.r.t. communication rounds (top row) and wall-clock time (bottom row) when training ResNet-18 image classification model using iNaturalist on Gaia underlay. 1 Gbps core links capacities, 100 Mbps access links capacities, $s = 5$.

(a) Train Loss      (b) Train Accuracy      (c) Test Loss      (d) Test Accuracy

Figure 21: Effect of overlays on the convergence w.r.t. communication rounds (top row) and wall-clock time (bottom row) when training ResNet-18 image classification model using iNaturalist on AWS North America underlay. 1 Gbps core links capacities, 100 Mbps access links capacities, $s = 5$.

(a) Train Loss      (b) Train Accuracy      (c) Test Loss      (d) Test Accuracy

Figure 22: Effect of overlays on the convergence w.r.t. communication rounds (top row) and wall-clock time (bottom row) when training ResNet-18 image classification model using iNaturalist on Géant underlay. 1 Gbps core links capacities, 100 Mbps access links capacities, $s = 5$.

(a) Train Loss  (b) Train Accuracy  (c) Test Loss  (d) Test Accuracy

Figure 23: Effect of overlays on the convergence w.r.t. communication rounds (top row) and wall-clock time (bottom row) when training ResNet-18 image classification model using iNaturalist on Exodus underlay. 1 Gbps core links capacities, 100 Mbps access links capacities, $s = 5$.

(a) Train Loss  (b) Train Accuracy  (c) Test Loss  (d) Test Accuracy

Figure 24: Effect of overlays on the convergence w.r.t. communication rounds (top row) and wall-clock time (bottom row) when training ResNet-18 image classification model using iNaturalist on Ebone underlay. 1 Gbps core links capacities, 100 Mbps access links capacities, $s = 5$.

## H.4 Training on Full-iNaturalist dataset

Full-iNaturalist contains 450,000 images belonging to 8142 classes. The distribution of images across classes is highly skewed. We randomly split them into an $80\%$ training set and a $20\%$ testing set, and fine-tuned a pretained ResNet-50 on ImageNet from torchvision implementation for species classification. When training on Gaia, AWS North America, and Géant networks, the initial learning rate is set to 5e-5 with Adam optimizer. When training on Exodus and Ebone networks, the initial learning rate is set to $0.1$ with SGD optimizer. We decay the learning rate by half every epoch. The batch size is set to $96$. Because of the larger model size ($161.06$ Mbits) and larger batch size (compared with the iNaturalist setting in Table 2), the computation time for one local update of the model in this case increases to $946.7$ ms.

Half of the images are assigned uniformly at random, the other half are assigned to the geographically closest silo. Table 8 shows that our method generates quite unbalanced data distributions (e.g., for Ebone, one silo can have up to 43 times more images than another one). Moreover, Figure 25 shows pairwise Jenson-Shannon (JS) divergence [63] for label distributions at different silos under our method and under a uniformly random repartition. The JS divergence across silos is larger when the samples are distributed following our method, suggesting that novel data is far from being iid distributed.

Table 8: Statistics of Full-iNaturalist dataset distribution for different networks.

| Network name | Silos | Samples/silo | | | |
| --- | --- | --- | --- | --- | --- |
| | | Mean | Stdev | Min | Max |
| Gaia | 11 | 37795 | 29986 | 19344 | 112745 |
| AWS North America | 22 | 18897 | 9915 | 10502 | 50727 |
| Géant | 40 | 10393 | 17535 | 5102 | 116498 |
| Exodus | 79 | 5262 | 3368 | 2710 | 18454 |
| Ebone | 87 | 4778 | 11222 | 2264 | 98886 |

(a) Gaia     (b) AWS NA     (c) Géant     (d) Exodus     (e) Ebone

Figure 25: Pairwise Jensen-Shannon divergence across silos labels distributions for Full-iNaturalist dataset on different networks. The first row is for data distributed with our method and the second row is for data distributed uniformly at random.

Differently from the previous experiments, we did not set the consensus weights using the local degree rule, but, for a given overlay, we computed the consensus matrix $A$ with the optimal spectral properties. For undirected topologies, we solved the symmetric fast distributed linear averaging problem [62, Eq. 17]. This problem is expressed as a semi-definite program that is convex and can be solved efficiently. For the RING, the optimal consensus matrix has all the non-zero entries equal to $1/2$.

Table 9 shows the effect of 6 different overlays when training ResNet-50 over Full-iNaturalist in networks with capacities equal to 1 Gbps for core links and access links.[15] We can see that RING always achieves the best throughput in this setting.

## H.5   Dependence of model performance on underlays

The models obtained by the experiments in Table 3 have different performance w.r.t. the underlays. The reason is that we chose to optimize the mean of local functions (1), which leads to different optimization problems when the number of silos changes. The observed difference in the trained models' performances is related to the fact that each of them is the result of a different optimization problem. Instead, when optimizing the weighted sum of local functions with weights equal to the percentage of the data points held by silos, the model performance does not depend on the underlay. To confirm this claim, we trained ResNet-18 on iNaturalist using the weighted average loss on STAR topology over the five underlays considered in the paper. Figure 26 shows that the obtained models for these five underlays have similar performances, reaching a test accuracy between $46\%$ and $48\%$.

## H.6   Effect of $C_b$ in MATCHA

There is no real configuration criterion for $C_b$ in [104], but [104, Fig. 3] suggests to select the smallest $C_b$ that has the same spectral norm of vanilla-SGD—but less communication overhead. This criterion leads to pick for all our topologies, but "AWS North America," a value of $C_b \in [0.4, 0.6]$, with no significant change to the results in Table 3. For "AWS North America" the criterion leads to $C_b = 0.2$.

Table 9: Full-iNaturalist training over different networks. 1 Gbps core links capacities, 1 Gbps access links capacities. One local computation step ($s = 1$).

| Network name | Silos | Links | Cycle time (ms) | | | | | Ring's training speed-up | |
|---|---|---|---|---|---|---|---|---|---|
| | | | STAR | MATCHA$^{(+)}$ | MST | $\delta$-MBST | RING | vs STAR | vs MATCHA$^{(+)}$ |
| Gaia [38] | 11 | 55 | 4444 | 2721 (2721) | 1498 | 1363 | **1156** | 3.84 | 12.10 (12.10) |
| AWS North America [96] | 22 | 231 | 7785 | 4384 (4384) | 1441 | 1297 | **1119** | 6.96 | 23.50 (23.50) |
| Géant [29] | 40 | 61 | 13585 | 4912 (1894) | 1944 | 1464 | **1196** | 11.35 | 4.10 (1.58) |
| Exodus [68] | 79 | 147 | 26258 | 6180 (1825) | 2078 | 1481 | **1194** | 13.74 | 2.59 (0.96) |
| Ebone [68] | 87 | 161 | 28753 | 8045 (1933) | 2448 | 1481 | **1178** | 19.52 | 5.80 (1.39) |

(a) Training loss vs Rounds

(b) Test accuracy vs Rounds

Figure 26: The model performance of training iNaturalist on STAR overlays of five different underlays: Gaia, AWS North America, Géant, Exodus and Ebone.

Table 10, first row, shows indeed that MATCHA is faster for $C_b = 0.2$, but still RING is $1.08$ and $3.29$ faster than MATCHA for 10 Gbps and 100 Mbps access links capacities, respectively. The table shows also that this criterion does not lead necessarily to the fastest training time for MATCHA. An alternative is to select $C_b$ by running time-consuming training experiments, but in any case we have always observed RING to outperform MATCHA except on Géant (see Table 3 and Table 10). Note that MATCHA is supposed to find by itself how often to use each link and "achieve a win-win in this error-runtime trade-off for *any arbitrary network topology*" [104]. We ran additional experiments with MATCHA over our topologies (for the RING we considered its undirected version as MATCHA uses bi-directional communications); however, MATCHA was still slower than RING (last two rows in Table 10).

Table 10: RING's training speed-up vs MATCHA when training iNaturalist on AWS-North America network. MATCHA runs on top of underlay, RING, and $\delta$-MBST with different values of communication budget $C_b$. 1 Gbps core links capacities. The star denotes the results with $C_b$ set according to [104, Fig. 3]. Bold fonts denote the optimal setting for $C_b$.

| Access links capacities | | 10 **Gbps** | | | | | | | 100 **Mbps** | | | | | | |
|---|---|---|---|---|---|---|---|---|---|---|---|---|---|---|---|
| **Communication budget ($C_b$)** | | 1.0 | 0.8 | 0.6 | 0.5 | 0.4 | 0.2 | 0.1 | 1.0 | 0.8 | 0.6 | 0.5 | 0.4 | 0.2 | 0.1 |
| **MATCHA over underlay** | | 2.02 | 1.43 | 1.57 | 1.47 | 1.46 | **1.08**$^*$ | 1.38 | 18.85 | 12.56 | 12.00 | 9.94 | 8.18 | 3.29$^*$ | **2.44** |
| **MATCHA over $\delta$-MBST** | | **1.10**$^*$ | 1.25 | 1.33 | 1.12 | 1.41 | 1.89 | 2.28 | 2.08$^*$ | 2.26 | 1.56 | 1.45 | 1.31 | **1.15** | 1.15 |
| **MATCHA over RING** | | **1.00**$^*$ | 1.42 | 1.40 | 1.15 | 1.26 | 1.35 | 1.34 | **1.00**$^*$ | 2.15 | 1.92 | 1.47 | 1.54 | 1.41 | 1.28 |