[Reviews · NeurIPS 2020]

Review 1

Summary and Contributions: In this paper, the authors propose a new approach to find the topology of underlying network of DPASGD with reduced cycle time. The authors define the minimal cycle time problem and solve it via approximation algorithms. Theoretical analysis and empirical results are provided.

Strengths: The authors define the minimal cycle time problem and solve it via approximation algorithms. Theoretical analysis is provided. Empirical results show the proposed algorithm can reduce the training time.

Weaknesses: Please correct me if I'm wrong. My major concern is that, according to the theoretical analysis of DPASGD and other decentralized optimization algorithms, the convergence depends on the eigenvalues of the connectivity graph. However, the proposed algorithms do not take the eigenvalues of the graph into account. Although the empirical results show good convergence, I believe the eigenvalue/graph Laplacian is a missing piece in the design of the MCT optimization problem and the theoretical analysis.

Correctness: The paper is technically sound.

Clarity: The paper is well written and easy to follow.

Relation to Prior Work: The difference and improvement compared to the previous work is clearly discussed.

Reproducibility: Yes

Additional Feedback: How does the topology affects the convergence of the decentralized optimization algorithm? Is there corresponding theoretical analysis? Is the convergence taken into account in the proposed algorithm? ----------------------- after authors' feedback After reading other reviews and authors' feedback, I agree that the convergence seems not significantly affected in these case. So the model in this paper makes sense. However, it will be very interesting if the authors could also consider the training convergence in future work.


Review 2

Summary and Contributions: This paper studies the optimal topology design problem in the setting of cross-silo federated learning. It nicely model the run time per iteration of decentralized SGD by using max-plus algebra and proposes several algorithms to obtain the overlay network in different scenarios.

Strengths: - The paper is well written and easy to follow. - The runtime modeling part is novel and valuable for the decentralized optimization community

Weaknesses: - Some claims are a bit questionable. I stated it in the "correctness" section - The authors seem to misunderstand MATCHA algorithm. I stated it in the "relation to prior work" section

Correctness: The claim that the convergence speed of decentralized SGD is weakly sensitive to the density of the overlay is questionable. I doubt this conclusion highly relies on the datasets and neural network models used in the experiments. For example, in [1], it is reported the final accuracy of a denser graph is consistently higher than a sparser one. Otherwise, there is no need to design the topology. Why not directly use the ring topology which is nearly the fastest and sparsest one? [1] stochastic gradient push for distributed deep learning. Assran et al. ICML 2019.

Clarity: Overall the paper is well written.

Relation to Prior Work: After checking the MATCHA paper, I feel the authors somehow misunderstand the MATCHA algorithm. It not only model the communication time in decentralized SGD but also can be used to identify and communicate more frequently over the important links in any fixed graphs. (This is why they optimize the algebraic connectivity) So one can also build MATCHA on the top of the overlay, which is the output of the MCT algorithm in this paper. Besides, the communication budget C_b in MATCHA can be tuned to balance throughput and final accuracy. It is unfair to fix C_b=0.5 in all experiments. The authors are supposed to make the above two points clear in the paper. It would be great to add more experiments on: (1) build MATCHA on the overlay (2) changing C_b in MATCHA.

Reproducibility: Yes

Additional Feedback: After rebuttal: (1) I have read the other reviewers' comments and the author's feedback. I agree that although this paper only focuses on the system aspect, the authors executed it very well. The communication time model developed in this paper may be valuable for future researchers. So I would like to recommend an acceptance. (2) I notice that the authors constraint the topology to be a strong digraph. Hence, it implicitly puts some requirements on the spectral properties of the graph. (3) I feel the tile 'optimal topology design' seems to be too general, given the fact that the authors didn't explicitly discuss the convergence properties of the algorithm. The authors should change the title to something like 'Cycle-time optimal topology design' or 'Communication-time optimal topology design'. (4) It would be better to analytically define edge-capacitated and node-capacitated networks. ----------------------------- This paper overall is a good and addresses an important problem. But the experimental results is kind of contradictory to what they are proposing. From figure 2, one can easily draw the conclusion that the ring topology is nearly the fastest. If the topology doesn't influence the convergence rate a lot, why do we need a complicated topology design algorithm as proposed by this paper to find denser topologies?


Review 3

Summary and Contributions: This work considers the setup of cross-silo federated learning and a question of how to design connection topology in order to minimize total wall clock time. Prior to reading this work, I was skeptical about the value of this line of work. This work, well motivated and bringing insights from areas I would not normally follow, certainly challenges my intuition. Overall, I would like to see this work accepted, but I am not sure how big of a problem is what I highlight regarding experimental evaluation, and my overall score could change both ways after rebuttal.

Strengths: I appreciate being clear what kind of federated learning setup this work applies to as well as when it is not relevant. The whole work is very well motivated and grounded in realistic considerations. In my view this work raises the bar for how to realistically setup simulated experiments relevant for the problem studied. Takes a stab at relatively clearly stated question in prior work with little attention. Relevant to a relatively new setup of increasing interest reported by multiple companies.

Weaknesses: While experimental setup is very good on the side of simulation of system properties, I feel it is weak on the algorithmic side. See below.

Correctness: I believe yes.

Clarity: Clearly written, easy to follow.

Relation to Prior Work: Very well grounded in recent works spanning multiple areas.

Reproducibility: Yes

Additional Feedback: I have read the response and other authors comments. I am glad to hear that optimizing for sum of local functions yields the same final accuracy independently from the network. I strongly recommend to redo all of the experiments with this objective, and feel confident it should work. This will make the experiments more conclusive, by having the results comparable across different networks via sharing the same objective. If the final convergence accuracies are very close, it also strongly addresses some other concerns, including mine, about how the topology impacts algorithmic convergence properties. Overall I feel this work exceeds the quality of a few other related and accepted works, and thus would really like to see it accepted, and have increased the rating. initial content --- Abstract: "master-slave" - I think there is a general trend in CS to move away from this language, you may want to consider "server-client" "parent-child" or something similar. In general the paper reads well, there are only minor details I have regarding clarity of presentation. I do appreciate being upfront about what kind of scenarios this work applies to and what not. Sec 1 I think very well grounds the rest of the work in prior research. In particular, in contrast to some related recent works, I feel this thorough grounding helps the authors to ask better questions, and in this sense I feel this work could help inspire further research. L65: I am not sure how compression plays into the preference for sychronous algorithms. It is relevant though, including for the precise topic studied here. Perhaps missing reference is Caldas et al., "Expanding the Reach of Federated Learning by Reducing Client Resource Requirements" which does both model and update compression as in the text. Sec 2.4. Here or earlier, I recommend spelling out that changing the overlay changes the algorithm executed, and thus potentially the convergence properties. The impact of this is studied in some previous works which you do cite and mention later on. I do not think this is a major problem, it is ok to focus on the system properties only, especially when this was not properly studied previously. Sec 3 formulates some practical approaches to finding the solution in prev section or its approximation, based on whether the communiction plays a major role or not. Sec 4 - I both like and do not like the experiments. I like that it is very well motivated and a lot of effort was clearly spent to create a novel simulated setup that should reflect realistic problems as much as possible, including using real topologies of internet infrastructure. This sets the bar higher for the whole field and will help followup research support more persuasive conclusions. I think this is very valuable for the research community and appreciated. What I do *not* like is stemming from footnote 5. The model learned should not be impacted by data partitioning and connectivity network. So measuring speed to reaching different thresholds seems wrong to me. I am not sure (and would like answered in rebuttal) is what is causing this. Two possible explanations I can imagine: a) Eq (1) is sum of averages. This way, a client with little data has disproportionate impact, possibly with negative impact on accuracy of the trained model. By default, I would expect overall average over data points, ususally structured as weighted average of local averages. That way, the data partitioning is irrelevant in terms of the global optimization objective. But maybe the outer average is forgotten in the equation? b) The default hyperparameters (say for STAR topology) are not set well. It may be that different topologies require different learning rates in order to converge, or to converge reasonably fast. Minor: In Fig 2 I would much more prefer to see a notion of test accuracy on the y-axis, not training loss. This way it is not clear at all whether the differences visualized translate to any difference in the task of interest. These are in the appendix, so a minor point. Minor: L235-236 I did not like the description of data partitioning (why is half of the data split randomly???). But the expanded reason in appendix make me feel ok about this. Consider expanding this aspect of the setup in the main body. L257: which table?

[Author Response · NeurIPS 2020]

**Effect of topology spectral properties on convergence (All Reviews)** Spectral properties of the topology do have an
effect on the number of iterations to converge (lines 42–44 in our paper). But there is evidence that, in practice, this effect
has been over-estimated by classic worst-case convergence bounds from [23,71,73]. We point the reviewers to [75],
which partially explains the phenomenon and overviews theoretical results proving asymptotic topology-independence
[58,81,5] and experimental evidence on image classification tasks ([75, Fig. 2], [62, Fig. 20], [58, Fig. 3]) and—we
add—natural language processing tasks [58, Figs. 13-16]. Reference [47, Sec. 6.3] extends some of the conclusions
in [75] to dynamic topologies and multiple local updates. Paper [5] (pointed in Review #2) proves convergence to be
asymptotically topology-insensitive [5, Th.1] and shows the topology has a more significant effect on system throughput
than on the number of iterations to converge [5, Figs. 1b and 3] with sparser topologies achieving higher accuracy *after*
*the same time* [5,Table 5]. Motivated by these observations, we design topologies to maximize throughput rather than
spectral gap (convergence is guaranteed by the choice of the consensus matrix). Our experiments show that this choice
is correct, as the topologies selected by our algorithms achieve faster training than the STAR, which has optimal spectral
properties, and MATCHA, which optimizes spectral properties given a communication budget (lines 299–301). As
suggested in Review #1, we will explore how to further improve speedup by taking into account the spectral properties
(lines 301–302). Our recent results show that, by optimizing the weights of the consensus matrix, MST training speedup
in comparison to the STAR increases on average by an additional 20% (e.g. from 6x to 7.2x for Géant).

**Why designing the topology if RING is nearly the fastest and sparsest one? (Review #2)** RING is not always the
fastest topology: MST and $\delta$-MBST may have a smaller cycle time (and then larger throughput) than RING (Fig. 3),
in particular when delays in the core network are the dominant component in Eq. (3). Whenever MST and $\delta$-MBST
have throughput close to RING, they achieve faster training, as they have better spectral properties. For example,
MST achieves $1.35$ training speed-up vs RING when training iNaturalist over Géant (not reported in Table 3). After
these clarifications, the fact that the RING is often the best choice is indeed a perhaps surprising finding of our paper.
Nevertheless, note that the RING we consider is not a generic ring (there are $(n-1)!$ rings in a complete $n$-node
network), but the one selected by our algorithm with provable guarantees on the cycle time (Prop. 3.3 and 3.6).

**Comparison with MATCHA (Review #2)** The reviewer is right that MATCHA [99] selects more frequently the
important links. Our description of MATCHA in the introduction may be too synthetic, but we confirm that we
implemented the complete MATCHA solution including the optimization of matching selection probabilities according
to (5) in [99] (see function `get_matching_activation_probabilities` in our code). For the communication
budget we have selected the value $C_b = 0.5$, which is the typical value used in experiments in [99]. There is no real
configuration criterion for $C_b$ in [99], but [99, Fig. 3] suggests to select the smallest $C_b$ that has the same spectral
norm of vanilla-SGD—but less communication overhead. This criterion leads to pick for all our topologies, but "AWS
North America," a value of $C_b \in [0.4, 0.6]$, with no significant change to the results in our paper—actually sometimes
MATCHA performs worse than before. For "AWS North America" the criterion leads to $C_b = 0.2$. Table 1, first row,
shows indeed that MATCHA is faster for $C_b = 0.2$, but still RING is $1.08$ and $3.29$ faster than MATCHA for 10 Gbps
and 100 Mbps access links capacities, respectively. The table shows also that this criterion does not lead necessarily to
the fastest training time for MATCHA. The alternative is to select $C_b$ by running time-consuming training experiments,
but in any case we have always observed RING to outperform MATCHA except on Géant (Table 1 below and Table 3
in our paper). Review #2 suggested to run MATCHA on our overlays. Note that MATCHA is supposed to find by itself
how often to use each link and "achieve a win-win in this error-runtime trade-off for *any arbitrary network topology*"
[99]. We run additional experiments with MATCHA over our topologies (for the ring we consider its undirected version
as MATCHA uses bi-directional communications); however, MATCHA is still slower than RING (Table 1).

**Impact of data partitioning on the model (Review #3)** The different accuracy reached in the different networks is
indeed due to minimizing the sum of the average losses as indicated in (1). There are two common settings in federated
learning [53, Eq. 1], either optimizing the mean of local functions [39, Eq. 2] as considered in our and MATCHA's
paper (arguably better for per-client fairness in non-IID settings), or optimizing the weighted sum of local functions
as suggested by Review #3. We run experiments also in the second case and confirm that models' final accuracy is
independent from the network. The other reviewer's comments are also spot on, we will improve the paper accordingly.

Table 1: RING's training speed-up vs MATCHA in AWS-North America network. MATCHA runs on top of underlay, RING, and $\delta$-MBST with different values of communication budget $C_b$. 1 Gbps core links capacities. The star denotes the results with $C_b$ set according to [99, Fig. 3]. Bold fonts denote the optimal setting for $C_b$.

| Access links capacities | | 10 **Gbps** | | | | | | | 100 **Mbps** | | | | | | |
|---|---|---|---|---|---|---|---|---|---|---|---|---|---|---|---|
| **Communication budget (C$_b$)** | | 1.0 | 0.8 | 0.6 | 0.5 | 0.4 | 0.2 | 0.1 | 1.0 | 0.8 | 0.6 | 0.5 | 0.4 | 0.2 | 0.1 |
| **MATCHA over underlay** | | 2.02 | 1.43 | 1.57 | 1.47 | 1.46 | **1.08**$^*$ | 1.38 | 18.85 | 12.56 | 12.00 | 9.94 | 8.18 | 3.29$^*$ | **2.44** |
| **MATCHA over $\delta$-MBST** | | **1.10**$^*$ | 1.25 | 1.33 | 1.12 | 1.41 | 1.89 | 2.28 | 2.08$^*$ | 2.26 | 1.56 | 1.45 | 1.31 | **1.15** | 1.15 |
| **MATCHA over RING** | | **1.00**$^*$ | 1.42 | 1.40 | 1.15 | 1.26 | 1.35 | 1.34 | **1.00**$^*$ | 2.15 | 1.92 | 1.47 | 1.54 | 1.41 | 1.28 |

[Meta-Review · NeurIPS 2020]

The paper proposes methods for designing communication graph for the decentralized periodic averaging SGD (DPASGD) in the federated learning set up focusing on reducing the per-iteration complexity (cycle time). The reviews were very appreciative of the good system and experimental design aspects of the paper that accounts for various types of delays in realistic scenarios. I would like to thank the authors for their effort. The reviewers were quite engaged and have provided many useful feedback and I hope these will be used to improve the paper. In particular, I would like to comment of few points -- please see full reviews for details - Although the authors motivate the need for focusing on cycle time over convergence rate in the introduction, based on the reviews, I believe it would be useful to include this discussion explicitly as a highlighted paragraph or subsection (see also comments by R2 on digraph constraint) - I would also encourage you to consider the title change suggestion by R2 (or something similar) as I and other reviewers agree that the current title is too generic. - technical terms like edge-capacitated and node-capacitated networks, access delay are used quite prominently but not precisely/formally defined. For a general audience and some in-field reviewers, this was confusing. Please organize these definitions more systematically. Relating to the above, the AC had the following question which was not clarified in the submission and I hope the authors will address in the final version. “My understanding is that in edge-capacitated networks the M/min(C_up/N_i,C_dn/N_j,A(i',j')) is ignored while in node capacitated networks they cannot be. This would mean that the MSR and RING topologies suggested in Sec 3.1 for edge-capacited networks should be optimal when bandwidth is *large* enough. However Fig 3 and the discussion seems to suggest that these algorithms are more effective for smaller bandwidths. The discussion in lines 270-275 also suggests that RING is faster for slower networks. Isnt this contradictory?”